


Modelling of street-scale pollutant dispersion by coupled simulation of
chemical reaction, aerosol dynamics, and CFD
Chao Lin [a,*,★], Yunyi Wang [b,★], Ryozo Ooka [c], Cédric Flageul [d], Youngseob Kim [b],
Hideki Kikumoto [c], Zhizhao Wang [b], Karine Sartelet [b]
[a] Graduate School of Engineering, The University of Tokyo, 4-6-1 Komaba, Meguro-ku,
Tokyo 153-8505, Japan
[b] CEREA, École des Ponts ParisTech, EdF R&D, 77 455 Marne la Vallée, France
[c] Institute of Industrial Science, The University of Tokyo, 4-6-1 Komaba, Meguro-ku,
Tokyo 153-8505, Japan
[d] Curiosity Group, Pprime Institute, Université de Poitiers, CNRS, ISAE-ENSMA,
Chasseneuil, France
[*] Corresponding author, c-lin415@iis.u-tokyo.ac.jp
★These authors contributed equally to this work.
ABSTRACT
In the urban environment, gas such as nitrogen dioxide $NO_2$, and particles impose adverse
impacts on pedestrians' health. The conventional computational fluid dynamics (CFD)
methods that regard pollutant as passive scalar cannot reproduce the formation of
secondary pollutants, such as $NO_2$ and secondary inorganic and organic aerosols, leading
to uncertain prediction. In this study, SSH-Aerosol, a modular box model that simulates
the evolution of gas, primary and secondary aerosols, is coupled with the CFD software
OpenFOAM and Code_Saturne. The transient dispersion of pollutants emitted from
traffic in a street canyon is simulated using unsteady Reynolds-averaged Navier–Stokes
equations (RANS) model.
The simulated concentrations of $NO_2$, $PM_{10}$ and black carbon are compared with field
measurements on a street of Greater Paris. The simulated $NO_2$ and $PM_{10}$ concentrations
based on the coupled model achieved better agreement with measurement data than the
conventional CFD simulation. Meanwhile, the black carbon concentration is
underestimated, probably partly because of the underestimation of non-exhaust emissions
(tyre and road wear).
Vehicles are considered the main source of ammonia ($NH_3$) in urban environments, which
may condense with nitric acid ($HNO_3$) to form ammonium nitrate. In the reference
simulation with $NH_3$ traffic emissions accounting for 1-2% of NOx emissions, aerosol
dynamics leads to an ammonium nitrate increase of 46% on average over a 12-hour
simulation period (5 a.m. to 5 p.m.) compared to the conventional CFD simulation.
Furthermore, an increase in $NH_3$ traffic emissions (to 10% and 20% of NOx emissions)





may leads to a large increase in ammonium nitrate (35% and 55%) compared to the
reference simulation.
In addition, aerosol dynamics leads to a 52% increase in 12-hour time-averaged organic
matter concentrations compared to the conventional CFD simulation, because of the
condensation of anthropogenic compounds from precursor-gas emissions and of
background biogenic precursor-gases on the enhance inorganic concentrations.
Keywords
Pollutant dispersion, Street canyon, Aerosol dynamics, CFD, $PM_{10}$, Secondary aerosols

**1. Introduction**

Traffic-related pollutants can impose adverse effects on pedestrians' health in the urban
environment (Jones et al., 2008; Anenberg et al., 2017). Especially, particulate matter
(PM) is strongly associated with increased cardiovascular diseases (Du et al., 2016).
Therefore, investigating the dispersion of PM and the corresponding precursor gas is of
great significance to evaluate the environmental impact and devise suitable
countermeasures (Kumar et al., 2008).
With the development of numerical simulations, computational fluid dynamics (CFD) has
been widely used for near-field dispersion prediction (Tominaga and Stathopoulos, 2013).
The pollutant dispersion patterns in complex geometric and non-uniform building
configurations can be well predicted using CFD simulations (Blocken et al., 2013).
Pollutant dispersion, deposition and transformation (chemical reactions and aerosol
dynamics) have primary roles in near-field prediction models. However, most CFD-based
studies assume that the time scale of transport at the street scale (~ 100 m) is relatively
shorter than the time scale of deposition and transformation; therefore, they frequently
regard pollutants as inert matter. Meanwhile, the recirculation flows which commonly
exist in street canyons lead to low-ventilation zones and may provide sufficient time for
transformation (Lo and Ngan, 2017; Zhang et al., 2020).
In addition, when PM is transported as a passive scalar, the distribution of the total
concentration can be simulated, however, information on the particle size distribution and
chemical composition is unclear. Understanding the size distribution is important for
evaluating the health hazards because large particles are deposited in the mouth and upper
airways, whereas smaller particles deposit deeper in the lungs and can even reach the
alveolar region of the lungs (Sung et al., 2007). In addition, as particles of different
chemical compositions are related to different sources and/or precursor gases, gaining
knowledge of their composition may help devise countermeasures to limit their



concentrations (Kim, 2019).
To simulate pollutant concentrations considering both transport and transformation, many
studies have coupled air-quality models with gas-phase chemistry and aerosol modules
and achieved chemical transport from a regional scale (~100 km) (Sartelet et al., 2007) to
a street scale (Lugon et al., 2021b). However, few models can simultaneously represent
detailed particle dispersion in a complicated urban flow field considering secondary
aerosol formation.
For the recent development and application of CFD-chemistry coupling model, Kurppa
et al. (2019) implemented a sectional aerosol module into large eddy simulation (LES),
and conducted a particle dispersion simulation on a neighborhood scale. Gao et al. (2022)
employed the same model to examine the dispersion of cooking-generated aerosols in an
urban street canyon. In both studies, the effect of particle dynamics on aerosol number
concentration was well reproduced. However, the simulated chemical composition was
not detailed. In addition, the chemical reactions of the precursor gas were not considered.
Kim et al. (2019) coupled unsteady Reynolds-averaged Navier-Stokes (RANS) model
with gas chemistry and aerosol modules and conducted simulations of $PM_1$ in a street
canyon under summer and winter conditions. The diurnal variations, spatial distribution
and chemical composition of pollutants in the street canyon were investigated. However,
the size distribution of particles and the secondary organic aerosol (SOA) chemistry were
not considered.
Therefore, to achieve a more comprehensive simulation of PM and related precursor gas,
this study coupled two open-source CFD softwares: OpenFOAM (OpenFOAM, 2020)
and Code_Saturne (Archambeau et al., 2004), with gas-phase chemistry and aerosol
module SSH-Aerosol (Sartelet et al., 2020). Simulations of the PM concentrations in a
two-dimensional street canyon are conducted. The coupled model is validated by
comparison to field measurements. The size distributions and chemical compositions of
particles from the models with and without secondary aerosol formation are compared.
Vehicles are considered the main ammonia ($NH_3$) source in urban environments (Sun et
al., 2017). Reactive nitrogen emissions from many new model year vehicles are now
dominated by $NH_3$ (Bishop and Stedman, 2015). Since the formation of ammonium
nitrate is often limited by $HNO_3$ rather than $NH_3$ in urban areas ($NH_3$-limited), increasing
$NH_3$ may lead to increased ammonium nitrate production and PM concentration in urban
streets (Lugon et al., 2021b). However, $NH_3$ emissions from passenger cars are usually
not regulated (Suarez-Bertoa and Astorga, 2018). Therefore, to provide evidence in
making policies for $NH_3$ emission regulation, it is important to investigate the local
influence of $NH_3$ emissions on PM increase. As an illustration, cases considering large



NH$_3$ emissions are considered and the related PM increase is investigated.
The remainder of this paper is organized as follows. The coupling of the aerosol model
and CFD is introduced in Section 2. The computational details are presented in Section 3.
In Section 4, the simulated pollutant concentrations are compared with field
measurements, followed by evaluations of the influence of the grid, coupling method and
time step. In Section 5, spatial and temporal variations in the concentrations are analyzed.
The chemical compositions and size distributions of the particles between the coupled
model and the model that does not consider gas chemistry or aerosol dynamics are
compared. In addition, the effect of NH$_3$ traffic emissions on particle concentrations is
discussed. Finally, the conclusions and perspectives are presented in Section 6.

**2. Model description**
OpenFOAM v2012 and Code_Saturne 6.2 were used to solve the governing equations of
the flow field and transport equations of gas and particle mass fractions. The unsteady
RANS model was used for the transient simulations with both CFD codes. In OpenFOAM,
the RNG k–ε model (Yakhot et al., 1992) is deployed for turbulence closure. All transport
equations are discretized using the total variation diminishing (TVD) scheme (Harten,
1984; Yee, 1987), which combines the first-order upwind difference scheme and the
second-order central difference scheme. The PIMPLE algorithm, a merged PISO–
SIMPLE algorithm in the OpenFOAM toolkit, was used for pressure–velocity coupling.
In Code_Saturne, turbulence was solved using the k–ε turbulence model (linear
production) (Guimet and Laurence, 2002). The time and space discretizations of velocity,
pressure and other scalars in all transport equations are realized through a centred scheme
and a fractional step scheme (Archambeau et al., 2004) . For both CFD software, the dry
deposition schemes for gas and particle are added to the transport equations using volume
sink terms based on Zhang et al. (2003) and Zhang et al. (2001), respectively. The details
of the implementation are provided in Appendix A.
SSH-Aerosol (Sartelet et al., 2020) is a modular box model that simulates the evolution
of not only gas concentrations but also the mass and number concentrations of primary
and secondary particles. In SSH-Aerosol 112 gas species and 40 particle species are
considered. The particle compounds are dust, black carbon, inorganics (sodium, sulphate,
ammonium, nitrate and chloride), primary organic aerosol (POA) and secondary organic
aerosol (SOA). Three main processes involved in aerosol dynamics (coagulation,
condensation /evaporation and nucleation) are included. The particle size distribution is
modelled using a sectional size distribution. In this study, nucleation was not considered
and six particle size sections were employed with bound diameters of 0.01, 0.04, 0.16,



0.4, 1.0, 2.5 and 10 μm.
The coupling between CFD and SSH-aerosol was achieved by using the application
program interface (API) of SSH-aerosol. The gas and particle concentrations were
initialized in CFD and are transported in the domain for each time step. For each grid
volume cell, these transported concentrations, as well as meteorological parameters, such
as temperature and humidity, are then sent to SSH-aerosol to advance one time step of
gaseous chemistry and aerosol dynamics. Once the SSH-aerosol calculation was
completed, the concentrations were sent back to the CFD for the next time step. The
influence of different operator splitting algorithms is discussed in Section 4.4.

**3. Simulation setup**

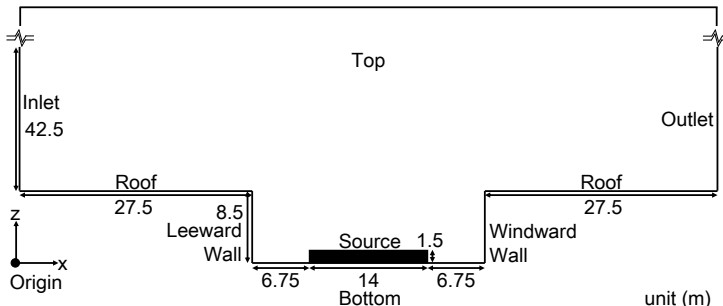


Fig. 1 Simulation domain of street canyon

The simulation was set up to model a street in Greater Paris (Boulevard Alsace-Lorraine),
where field measurements were conducted from April 6 to June 15, 2014. The
concentrations of nitrogen dioxide ($NO_2$), particles with diameters less than 10 μm ($PM_{10}$),
and black carbon were measured as described in Kim et al. (2018). Fig. 1 shows the
simulation domain. The 2-D street canyon is 27.5 m in width ($W$) and 8.5 m in height
($H$). The domain height was 6 H. The street canyon was discretized into uniform grids in
$x$- and $z$- directions. An analysis of the grid sensitivity is described in Section 4.3.
Simulations were conducted from 4:30 a.m. to 5 p.m. on April 30, 2014 at local time
(GMT+2). This period was selected because the wind direction was almost perpendicular
to the street canyon during that day, allowing for a 2D simulation setting. The first 30
minutes of the simulation corresponded to model spin-up, and the simulation lasted 12
hours. A sensitivity analysis of numerical aspects, such as the splitting method between
transport and chemistry and the time step, is described in Section 4.4.



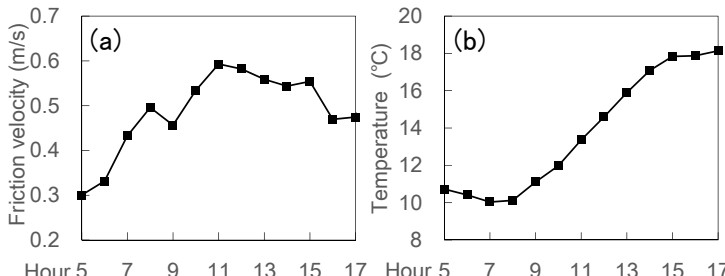

Fig. 2 Time variations of hourly friction velocity and temperature for inflow

Meteorological conditions (Fig. 2) including time-varying friction velocity and temperature were obtained from the simulation described in Sartelet et al. (2018) using the Weather Research and Forecasting (WRF) model. The lowest and highest friction velocities occurred approximately at 5 a.m. and 11 a.m., respectively. The lowest and highest temperatures occurred around 8 a.m. and 5 p.m. For the inflow, the wind direction was perpendicular to the street canyon. The friction velocity $u_*$ is used to prescribe the vertical profiles of the streamwise velocity $U$, turbulent kinetic energy $k$ and turbulent dissipation rate $\varepsilon$ as follows

$$U(z) = \frac{u_*}{\kappa} \ln\left(\frac{z - H}{z_0}\right) \tag{1}$$

$$k(z) = \frac{u_*^2}{\sqrt{C_\mu}} \tag{2}$$

$$\varepsilon(z) = \frac{u_*^3}{\kappa(z - H)} \tag{3}$$

where $\kappa$ is the von Kármán constant and $C_\mu$ is the model constant (=0.09) in the k-ε model. The roughness length $z_0$ is set to 1 m for the inlet (Belcher, 2005) and 0.1 m for the wall and bottom (Lo and Ngan, 2015).

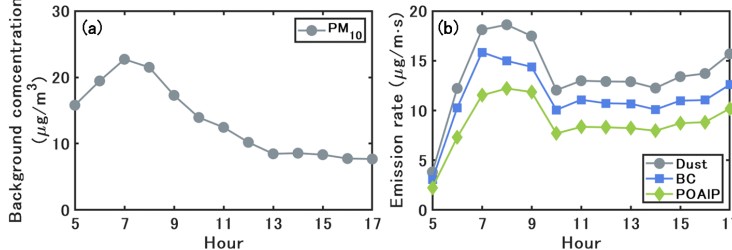

Fig. 3 Time variations of PM$_{10}$ background concentration (left panel) and emission rates



of dust, BC and organics (POAlP) (right panel).

Fig. 3 shows the time variations of the $PM_{10}$ background concentrations and emission
rates for the emitted compounds of $PM_{10}$. The background concentrations of the gas and
particles are obtained from the regional-scale simulations of Sartelet et al. (2018). The
hourly background concentrations were linearly interpolated into seconds and prescribed
at the inflow and top. The traffic emission source is assumed to be approximately 14 m
in width and 1.5 m in height, and it is set in the middle of the bottom of the canyon (Fig.
1). As detailed in Kim et al. (2022), emissions are estimated from the fleet composition
and the number of vehicles in the street using COPERT's emission factors (COmputer
Program to calculate Emissions from Road Transport, version 2019, EMEP/EEA, 2019).
After the speciation of $NO_x$, Volatile Organic Compounds (VOC), $PM_{2.5}$ and $PM_{10}$ into
model species, emissions were set for 16 gaseous model species and three particle model
species: dust and unspecified matter (Dust), black carbon (BC) and primary organic
aerosol of low volatility (POAlP).
For the boundary conditions of the OpenFOAM, the pressure and the gradients of all other
variables were set to zero at the outlet. For the walls, we used the wall functions of $\varepsilon$ and
turbulent kinematic viscosity $\nu_t$ for atmospheric boundary layer modelling in
OpenFOAM toolkit (OpenFOAM, 2020) based on Parente et al. (2011). The gradients of
turbulent kinetic energy $k$, concentration, and temperature were set to zero. In
Code_Saturne, a two-scales logarithmic friction velocity wall function was used for
solving the fluid velocity near wall cell and a three layers wall function is used for
computing other transported scalar profiles such as temperature near the wall (Arpaci and
Larsen, 1984).
The turbulent Schmidt number $Sc_t$ in the concentration transport equations, which is the
ratio of the turbulent diffusivity to the concentration and turbulent kinematic viscosity, is
important in turbulent diffusion modeling. The value of $Sc_t$ is considered between 0.2
and 1.3, depending on the flow properties and geometries (Tominaga and Stathopoulos,
2007). For urban environments with a compact layout, a small $Sc_t = 0.4$ is found to
show better agreement with wind tunnel experiment data (Di Sabatino et al., 2007).
Therefore, a value of 0.4 is adopted in the current study.

**4. Model evaluation**
*4.1. Validation with field measurements and comparison of simulated concentrations with*
*the two CFD software*

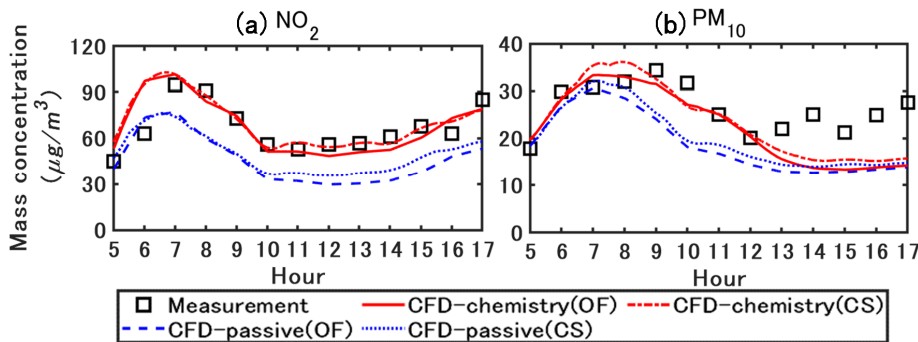


Fig. 4 Measured and simulated $NO_2$ and $PM_{10}$ concentrations. The values are spatially
averaged in the street canyon ($27.5 \leq x \leq 55, 0 \leq z \leq 8.5$ m). CFD-passive and CFD-
chemistry denote the CFD simulation without and with chemistry coupling. OF and CS
denote the simulated concentrations based on OpenFOAM and Code_Saturne. All
concentrations are represented in local time (GMT+2).

Fig. 4 compares the simulated concentrations with those obtained from the field
measurements. In this section, the results and discussion are based on the spatially-
averaged values in the street canyon ($27.5 \leq x \leq 55, 0 \leq z \leq 8.5$ m). CFD-passive and
CFD-chemistry denote the CFD simulation without and with chemistry coupling. OF and
CS denote simulated concentrations based on OpenFOAM and Code_Saturne.    The
operator splitting order and time step for OF and CS are the Strang method with 0.5 s and
the first order method with 0.25 s, as detailed in Section 4.4.
For $NO_2$, the peak concentration in the field measurement occurred approximately at 7
a.m. owing to the morning traffic. In the CFD-passive simulations, the lack of chemical
reactions lead to an underestimation of $NO_2$, while the concentrations simulated with
CFD-chemistry agree well with the measurements. For $PM_{10}$, the concentrations
simulated with CFD-chemistry also show better agreement with the measurements than
CFD-passive. The primary reason is that CFD-chemistry can reproduce the condensation
of inorganic and organic matters from the gas phase to the particle phase, which will be
further explained in the following sections. The simulation results based on OF and CS
show small differences, and detailed comparisons are presented in Fig. 6.
Validation metrics (Chang and Hanna, 2004) were used to quantify the overall accuracy
of the CFD simulated concentrations based on OF, compared with the measured values
(Ferrero et al., 2019; Trini Castelli et al., 2018). The following metrics were used:
fractional bias (FB), geometric mean bias (MG) and normalized mean square error





(NMSE). These metrics are defined as follows:

$$\text{FB} = \frac{\overline{Obs} - \overline{CFD}}{0.5(\overline{Obs} + \overline{CFD})} \tag{4}$$

$$\text{MG} = \exp(\overline{\ln Obs} - \overline{\ln CFD}) \tag{5}$$

$$\text{NMSE} = \frac{\overline{(Obs_i - CFD_i)^2}}{\overline{Obs} \times \overline{CFD}} \tag{6}$$

where $Obs_i$ and $CFD_i$ are the measured and CFD simulated concentrations for the
compound/species $i$, respectively. The overbar represents the mean value of the entire
dataset. The ideal values are 1 for MG, and 0 for FB and NMSE. Previous research has
suggested that $|\text{FB}| < 0.3$, $0.7 < \text{MG} < 1.3$ and $\text{NMSE} < 4$ are acceptable for
simulated concentrations (Hanna et al., 2004).
Table 1 shows the statistical indicators for spatially averaged concentrations of $NO_2$ and
$PM_{10}$ in the street canyon from 5 a.m. to 5 p.m. For $NO_2$ and $PM_{10}$, the mean and 90%
percentile concentrations simulated with CFD-chemistry are closer to the measurements
than those simulated with CFD-passive. In addition, the FB, MG and NMSE values of
CFD-chemistry are closer to the ideal values than those of CFD-passive.
Table 1 Statistical indicators for $NO_2$ and $PM_{10}$ in the street canyon from 5 a.m. to 5p.m.
The concentrations are simulated with OpenFOAM.

| | Concentration (μg/m³) | | Validation metrics | | |
|---|---|---|---|---|---|
| NO₂ | Mean | Percentile 90% | FB | MG | NMSE |
| Measurement | 66.6 | 91.8 | / | / | / |
| CFD-chemistry | 67.3 | 97.3 | -0.01 | 1.00 | 1E-4 |
| CFD-passive | 45.9 | 73.7 | 0.36 | 1.50 | 0.14 |
| PM₁₀ | Mean | Percentile 90% | FB | MG | NMSE |
| Measurement | 26.4 | 32.5 | / | / | / |
| CFD-chemistry | 22.3 | 33.1 | 0.17 | 1.23 | 0.03 |
| CFD-passive | 18.8 | 28.9 | 0.34 | 1.45 | 0.13 |




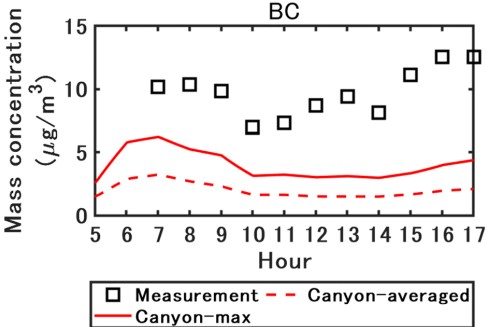


Fig. 5 Measured and simulated black carbon concentrations with OpenFOAM. The
canyon-averaged and maximum concentrations in the street canyon are represented by
the plain line and the dashed line respectively ($27.5 \leq x \leq 55, 0 \leq z \leq 8.5$ m).

The black carbon (BC) concentration simulated with OF was compared with the
measurements in Fig. 5. Because BC is considered an inert matter, considering chemistry
does not influence the mass concentration. Therefore, the concentrations simulated with
CFD-passive and CFD-chemistry show little difference; only the concentration simulated
with CFD-chemistry is shown here. The BC concentrations were underestimated by a
factor of approximately 5. Even the maximum concentrations in the street canyon largely
underestimate the measurements. One of the causes of this underestimation may be the
underestimation of the non-exhaust tyre emission factors in the COPERT emission factors
used here (Lugon et al., 2021a).

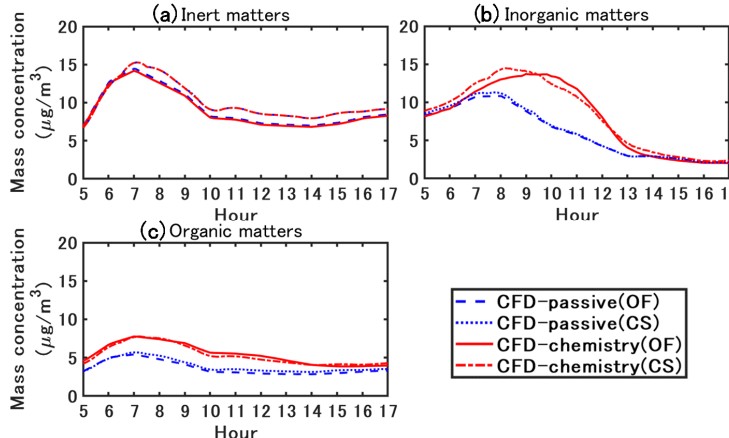


Fig. 6 Simulated particle concentrations with OpenFOAM (OF) and Code_Saturne (CS).
CFD-passive and CFD-chemistry denote the CFD simulation without and with chemistry





coupling.

The particle concentrations simulated with OF and CS are compared in Fig. 6. The
evolutions of the concentrations simulated by OF and CS were similar. Higher $PM_{10}$
concentrations were simulated by CS around 8 a.m. during the traffic peak and in the
afternoon, mostly because of the higher concentrations of emitted inert compounds, such
as black carbon and dust. Differences in the turbulence scheme may explain these
variations. Meanwhile, the difference between CFD-passive and CFD-chemistry for the
inorganic and organic matters was in accordance with OF and CS, showing the robustness
of the coupling method between CFD and SSH-aerosol by API. For simplicity, only the
simulated concentration based on OF is presented and discussed in the following sections.

*4.2. Transient-condition method and constant-condition method*

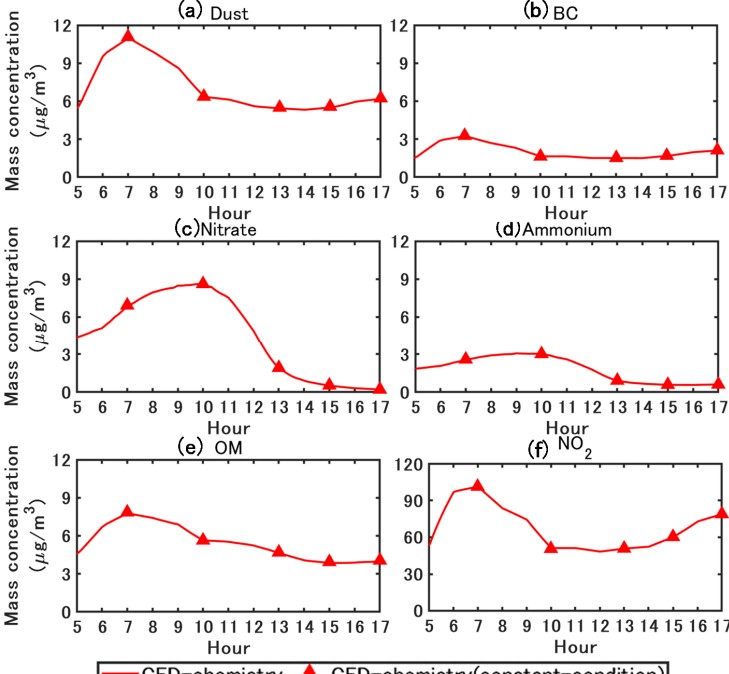


Fig. 7 Simulated $PM_{10}$ and $NO_2$ concentrations with the transient-condition and constant-
condition methods. The concentrations are spatially averaged in the street canyon.

Because time-varying concentrations are expensive to compute in terms of computational
time, conducting CFD simulations with fixed boundary conditions and emission rates at



specific time points is considered a practical method for evaluating street-level pollutant
concentrations (Wu et al., 2021; Zhang et al., 2020). The transport (advection and
diffusion) and chemical processes reached equilibrium, and the simulated concentrations
reached quasi-stable values. These values are often regarded as time-averaged
concentrations. This method is called the constant-condition method (CCM) in this study,
in contrast to the transient-condition method (TCM). However, the simulation accuracy
of CCM has not been validated in simulations that consider both gas chemistry and
particle dynamics. Therefore, validation was conducted using boundary conditions and
emission rates at five time points (7 a.m., 10 a.m., 1 p.m., 3 p.m. and 5 p.m.). Other
simulation conditions, including the grid, coupling method, and time step, are the same
as the transient-condition simulation.
In Fig. 7, for $PM_{10}$ and $NO_2$, the concentrations simulated with CCM (red triangles) were
similar to those simulated with TCM. In addition, depending on the background
concentration and emission conditions, the simulation time required for CCM to reach
dynamic equilibrium is less than 1000 time steps (approximately 500 s). Therefore, CCM
can be utilized for parameter studies. The sensitivity analysis of the grid, coupling method
and time step in Section 4.3 and 4.4 is based on CCM. However, it should be noted that
CCM cannot replace TCM when simulating long periods because the mass concentration
may not change linearly between the selected time points. The simulated concentrations
in Section 5 were based on TCM.
*4.3. Grid sensitivity*

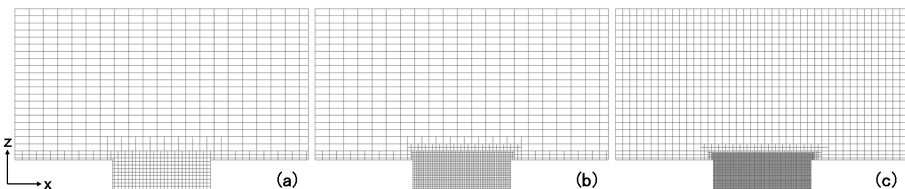

Fig. 8 Different grid resolutions for sensitivity analysis: (a) coarse, (b) basic, (c) fine. The
grid resolutions in the street canyon are 1 m, 0.5 m and 0.25 m in both $x$-and $z$- directions,
respectively. The largest grid sizes are 4 m (x) × 2m (z) in the coarse and basic grids,
and 2 m (x) × 2m (z) in the fine grid.
Grid sensitivity analysis was conducted based on three different resolutions as shown in
Fig. 8. The grid resolutions in the street canyon for coarse, basic and fine grids are 1 m,
0.5 m and 0.25 m in both $x$-and $z$- directions, respectively. The largest grid sizes are 4 m





(x) × 2m (z) for the coarse and basic grids, and 2 m (x) × 2m (z) for the fine grid. The
simulations were based on constant-condition method. The Strang method, which is
introduced in Section 4.4, is used with a time step of 0.5 s. Fig. 9 shows the comparative
results for the mass concentration. No significant discrepancy was observed between the
different grids for $NO_2$, inert matters and organic matters. Meanwhile, the simulated
inorganic matters based on coarse grids showed slightly smaller concentrations than the
other grid resolutions, while the concentrations based on basic and fine grids are close.
Therefore, the basic grid was adopted for simulations in this study.

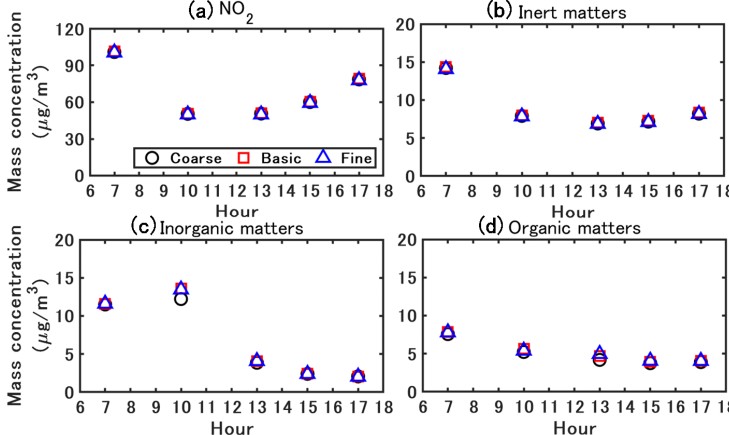


Fig. 9 Simulated $NO_2$ and particle concentrations with different grid resolutions.

*4.4. Coupling method and time step sensitivity*
The transport equation for the chemical species includes terms of advection, diffusion,
emission and chemical reactions. Ideally, the transport equation should be solved with all
the above terms, that is, by coupling all processes. However, the chemical process is
integrated with a stiff integrator, whereas advection, diffusion and emission are integrated
with a flux scheme. Therefore, operator splitting (Sportisse, 2000) is often employed to
solve different terms individually and sequentially over a given time step in chemical
transport simulations (Fu and Liang, 2016).
In this study, advection, diffusion and emission were simultaneously solved in CFD, and
the chemical reactions including gas chemistry, particle dynamics and size redistribution
were solved in SSH-Aerosol. Two integration orders are considered for coupling: a first
order method and a Strang method. For the first order method, which can be summarized
as CFD($\Delta t$)-Chemistry($\Delta t$), the mass concentrations are first integrated for transport over
a time step $\Delta t$. The updated concentrations are then integrated for chemistry at the





same $\Delta t$. This method is first-order accurate in time. To improve accuracy, Strang (1968)
introduced a symmetric sequence of operators, which can be summarized as CFD($\Delta t$/2)-
Chemistry($\Delta t$)-CFD($\Delta t$/2). The mass concentrations are first integrated for transport over
a half time step, then for chemistry over the full time step and finally for transport again
over a half time step. The Strang method leads to a second-order accuracy in time.
Table 2 Relative change in the computation time with different operator-splitting order
and time steps. The computation time is normalized by S05.

| Case | Operator splitting order | $\Delta t$ (s) | Change in the computation time |
|---|---|---|---|
| F05 | First order method CFD($\Delta t$)-Chemistry($\Delta t$) | 0.5 | 0.90 |
| F025 | | 0.25 | 1.56 |
| S1 | Strang method CFD($\Delta t$/2)-Chemistry($\Delta t$)-CFD($\Delta t$/2) | 1 | 0.57 |
| S05 | | 0.5 | 1 |
| S025 | | 0.25 | 2.44 |

A sensitivity analysis was conducted on the operator splitting method and splitting time
step. As shown in Table 2, the time step is considered 0.5 s and 0.25 s for the first order
method (named F05 and F025), and 1 s, 0.5 s and 0.25 s for the Strang method (named
S1, S05 and S025). The simulated $NO_2$ and particle concentrations are presented in Fig.
10. S1 and F05 concentrations hardly differed from the figures. Meanwhile, the
computational time of S1 was only 63% of that of F05. Similarly, the concentrations
simulated with S05 and F025 were almost the same, and the computational time of S05
was only 64% of F025. Therefore, the Strang method can be considered as a cost effective
method.

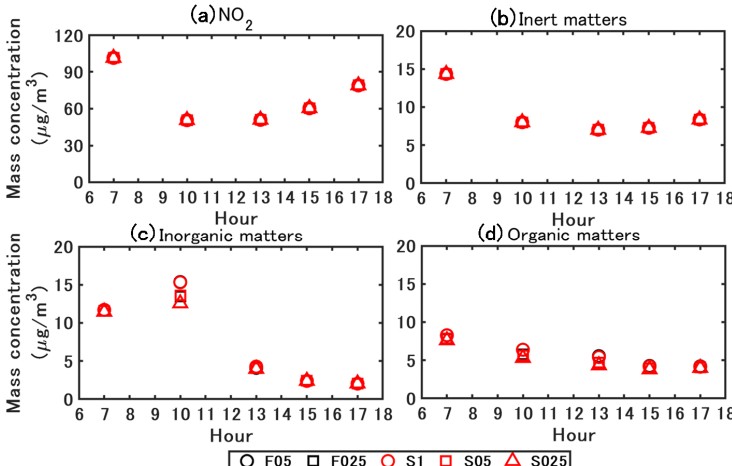


Fig. 10 Simulated NO$_2$ and particle concentrations with different coupling methods and time steps. S denotes the Strang method: CFD($\Delta t$/2)-Chemistry($\Delta t$)-CFD($\Delta t$/2). F denotes the first order method: CFD($\Delta t$)-Chemistry($\Delta t$). In the legend, the values that follow the capital letter S (Strang) or F (First order) denote the time step $\Delta t$ (in s) used in the simulation.

383

The concentrations simulated with the Strang method and different time steps show that small time step results in low inorganic and organic matter concentrations. The concentrations simulated with S1 were larger than those of S05, and larger than S025. However, the differences between the concentrations simulated with S05 and S025 were lower than the differences between S1 and S05. For NO$_2$ and inert particles, no obvious difference was found between the simulations with different splitting methods and splitting time steps. Therefore, the Strang method with a time step of 0.5 s is adopted in this study.

392



## 5. Results and discussion

*5.1. Time-averaged flow field and concentration field*

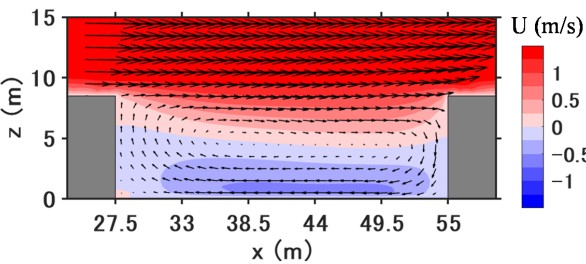

Fig. 11 Time-averaged flow field in the street canyon from 5 a.m. to 5 p.m.

This section shows the results for time-averaged values from 5 a.m. to 5 p.m. Fig. 11 shows the 12-hour time-averaged streamwise velocity and wind direction in the street canyon. At the current aspect ratio ($H/W$=0.31), a large vortex was observed in the canyon with a small secondary vortex at the corner of the leeward wall. A reverse flow was observed in the lower half of the canyon.

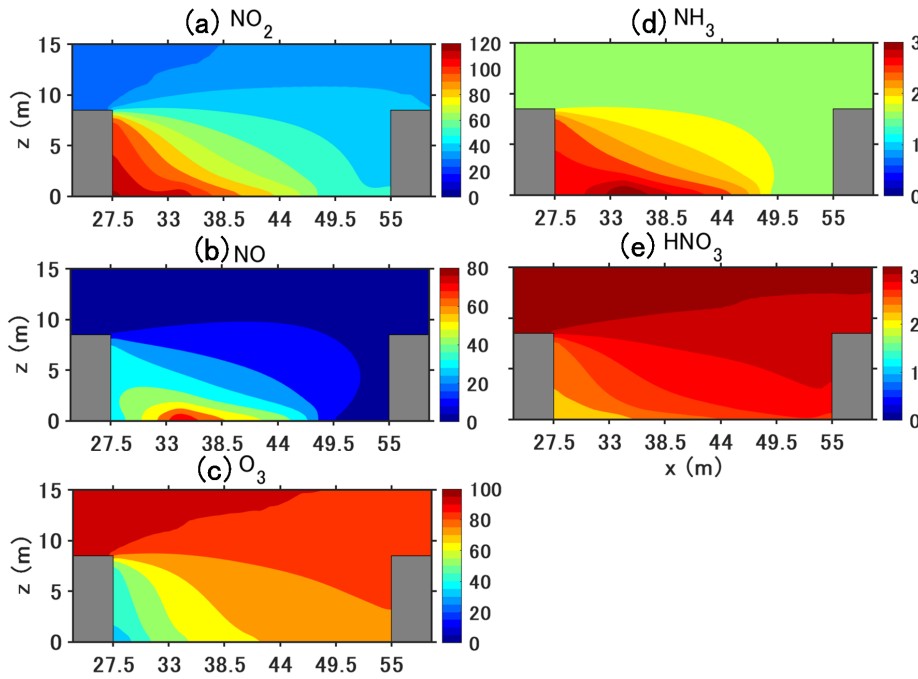

Fig. 12 Time-averaged concentrations (μg/m³) of gaseous pollutants in the street canyon





from 5 a.m. to 5 p.m.

Fig. 12 shows the time-averaged concentrations of the gaseous pollutants from 5 a.m. to
5 p.m. For gaseous pollutants emitted by traffic, such as $NO_2$, NO and $NH_3$, larger
concentrations are found in the street, particularly near the leeward wall, compared to the
windward wall due to the reverse flow. Simultaneously, gas-phase chemistry and
condensation/evaporation between the gas and particle phases also influence the
concentration distribution. $NO_2$ mainly increased due to chemical production from NO
emissions and background $O_3$. Compared to the background $NO_2$ concentration of 26
$\mu g/m^3$, the longest retention time at the leeward side corner led to the street canyon's
largest concentration (121 $\mu g/m^3$). At pedestrian height ($z$=1.5 m), $NO_2$ concentration was
116 $\mu g/m^3$ at the leeward wall and 49 $\mu g/m^3$ at the windward wall.
However, NO and $NH_3$ generally decreased because of loss by gaseous chemistry and the
condensation of ammonium nitrate, respectively; therefore, the largest concentrations
were at the leeward corner of the traffic emission source. For secondary gaseous
pollutants without traffic emissions such as $O_3$ and $HNO_3$, gaseous chemistry and
condensation led to lower concentrations near the leeward wall than background
concentrations. For $O_3$, this is due to the titration of $O_3$ by NO, whose concentration was
large near the leeward wall. For $HNO_3$, this was because of the high concentrations of
$NH_3$, which then condensed with $HNO_3$ to form ammonium nitrate.

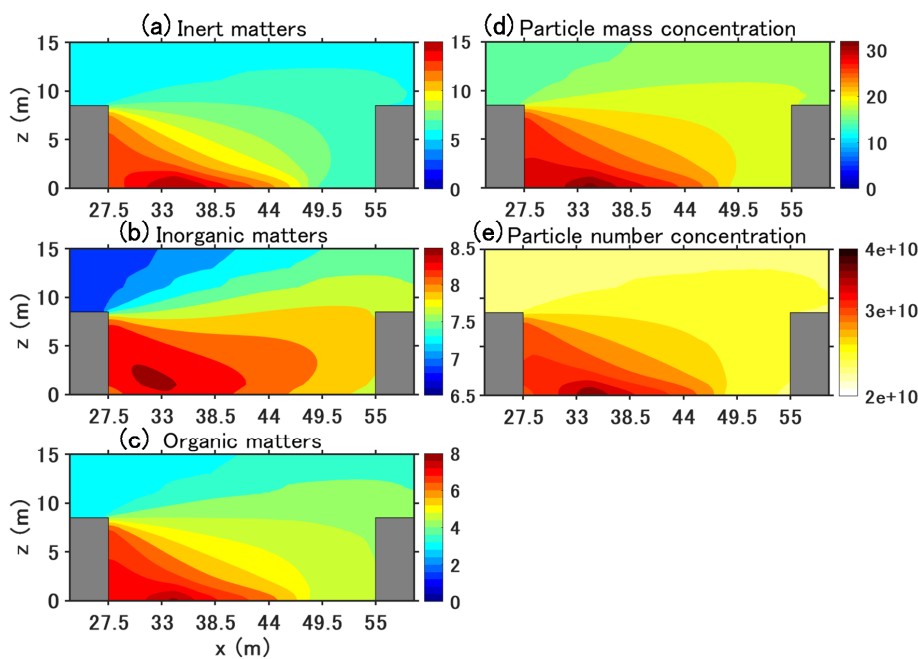

Fig. 13 Time-averaged concentrations of particle number, mass and composition in the
street canyon from 5 a.m. to 5 p.m. The unit is µg/m$^3$ for mass concentration and m$^{-3}$ for
number concentration.
Fig. 13 shows the time-averaged PM$_{10}$ mass concentration, and the number
concentrations and PM composition (inorganic, organic and inert matters) from 5 a.m. to
5 p.m. For inert and organic matters, the highest concentrations were near the leeward
corner of the traffic emission source. Because inorganic matters are not emitted, the
concentration distribution differs from inert and organic matters. However, as they are
produced from gas condensation and strongly influenced by traffic emissions, the highest
concentrations were observed in the leeward corner.
At pedestrian height ($z$=1.5 m), the PM$_{10}$ mass concentration was approximately 28 µg/m$^3$
at the leeward wall and 19 µg/m$^3$ at the windward wall, which is larger than the
background concentration of 15 µg/m$^3$. The number concentration is computed from the
mass concentration and therefore has a similar spatial distribution as PM$_{10}$ mass
concentration (nucleation from gas was not taken into account). Traffic emission
significantly increased the number concentration. The number concentration is about
$2.3 \times 10^{10}$ m$^{-3}$ in the background, whereas the largest number concentration in the
street canyon is about $3.8 \times 10^{10}$ m$^{-3}$.




*5.2. Time-variant characteristics*

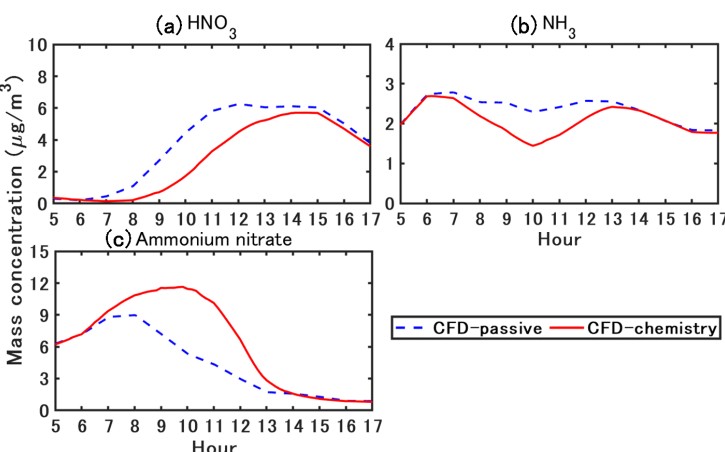


Fig. 14 Simulated time-varying concentrations of ammonium nitrate and precursor gas
(HNO$_3$ and NH$_3$).


Fig. 14 shows the simulated time-varying concentrations of ammonium nitrate formed by
the condensation of HNO$_3$ and NH$_3$. Based on the traffic fleet in the current study, NH$_3$
emission was approximately 1-2% of NOx emissions. Ammonium nitrate and HNO$_3$ are
not emitted and differences between simulations with or without chemistry coupling are
due to gas chemical reactions and phase change between the gas and particle. Phase
change may be driven by NH$_3$ emissions, as well as the non-thermodynamic equilibrium
of the background concentrations.
In CFD-passive, NH$_3$ concentration peaked around 6 am as NOx because it was emitted
by traffic. The peak in HNO$_3$ concentration was later in the morning, around 11 am. HNO$_3$
is formed from the oxidation of NO$_2$, which is emitted by traffic and is rapidly formed
from NO traffic emissions. The formation of HNO$_3$ is slower than the formation of NO$_2$,
and probably occurs at the regional scale, leading to a delay in the peak of HNO$_3$
concentration compared to NO$_2$ concentration. In CFD-chemistry, the temporal variations
of HNO$_3$ concentration show large differences with CFD-passive because HNO$_3$
condenses with NH$_3$ to form ammonium nitrate during the daytime. As a result, the HNO$_3$
concentration peak in CFD-chemistry was later than that in CFD-chemistry (it was shifted
from 11 a.m. to around 2 p.m.). The NH$_3$ concentration in CFD-passive peaked at 7 a.m.
because of traffic emission and was stable from 7 a.m. to 1 p.m. and then decreased from
1 p.m. Meanwhile, the condensation in CFD-chemistry leads to lower concentration than





in CFD-passive during the daytime (between 7 a.m. and 1 p.m.).
For 12-hour time-averaged concentrations, ammonium nitrate increased by 46% in CFD-
chemistry compared with that in CFD-passive. Background ammonium nitrate
concentration (CFD-passive) peaked around the morning rush (7 to 8 a.m.) and then
decreased. Meanwhile, in CFD-chemistry, ammonium nitrate concentration peaked later
around 10 a.m., because of the large increase in $HNO_3$ between the traffic rush and 10
a.m. However, although $HNO_3$ concentration did not vary much between 11 a.m. and 3
p.m., the ammonium nitrate concentration decreased from 10 a.m. to a very small level
(lower than 1 $\mu g/m^3$) after 2 p.m. This decrease is probably linked to the temperature
increase during the daytime (Fig. 2(b)) and the relative humidity decrease, leading to a
decrease in the condensation rate (Stelson and Seinfeld, 1982).

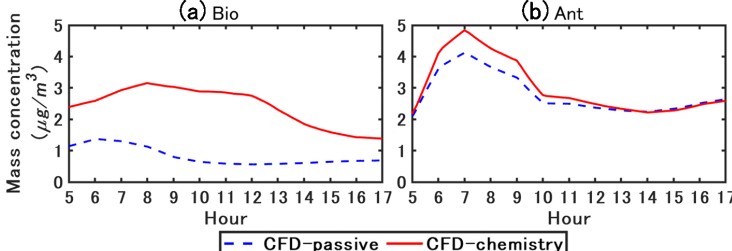


Fig. 15 Simulated time-varying concentration of organic matters. Bio refers to organic
matter formed from biogenic precursors. Ant refers to organic matter formed from
anthropogenic precursors.

Fig. 15 shows the simulated time-varying concentrations of organic matters. Organic
matters are divided into two main categories depending on the origin of the precursors:
Bio and Ant. refer to the organic matter of biogenic and anthropogenic precursors
respectively.
In CFD-chemistry, Bio concentration is larger than that in CFD-passive. As biogenic
precursors are not emitted in the street, the condensation of Bio is due to background
precursor gases. As discussed previously, the concentration of ammonium nitrate is higher
in CFD-chemistry than in CFD-passive, providing a larger aqueous mass onto which
hydrophilic compounds of the biogenic precursor gases condense. As the condensation of
ammonium nitrate decreased in the afternoon as shown in Fig. 14, the condensation of
Bio also decreased.
Ant is largely influenced by traffic emissions in the street, particularly by emissions of
semi-volatile compounds (Sartelet et al., 2018), which soon condense after emissions.
Therefore there is a peak around 7 a.m. owing to the morning rush. In the model,





anthropogenic emissions are mostly hydrophobic, therefore the condensation is not
enhanced by the increase in inorganic concentrations. Consequently, the difference
between CFD-chemistry and CFD-passive is larger in the morning owing to the large
increase in traffic emissions, but small differences are observed in the afternoon.

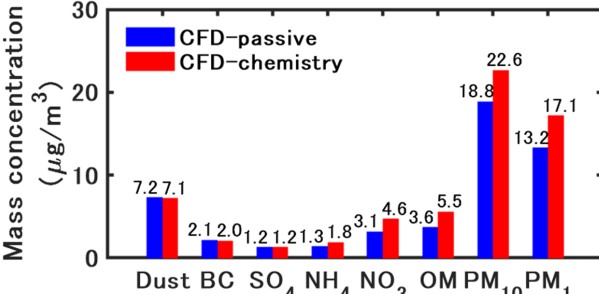

Fig. 16 Time-averaged concentration of the chemical compounds of $PM_{10}$, $PM_{10}$ and $PM_1$
from 5 a.m. to 5 p.m.

Fig. 16 shows the time-averaged concentrations of the chemical compounds of $PM_{10}$,
$PM_{10}$ and $PM_1$ from 5 a.m. to 5 p.m. The time-averaged $PM_{10}$ and $PM_1$ concentrations
increased by approximately 3.8 $\mu g/m^3$ in CFD-chemistry compared to CFD-passive,
indicating that chemistry mainly influences small particles. Inert matters slightly decrease
in CFD-chemistry owing to dry deposition. Condensation increases of 48%, 38% and
53% of nitrate, ammonium and organic matter concentrations in CFD-chemistry
compared to CFD-passive.



*5.3. Size distribution of particle matters*

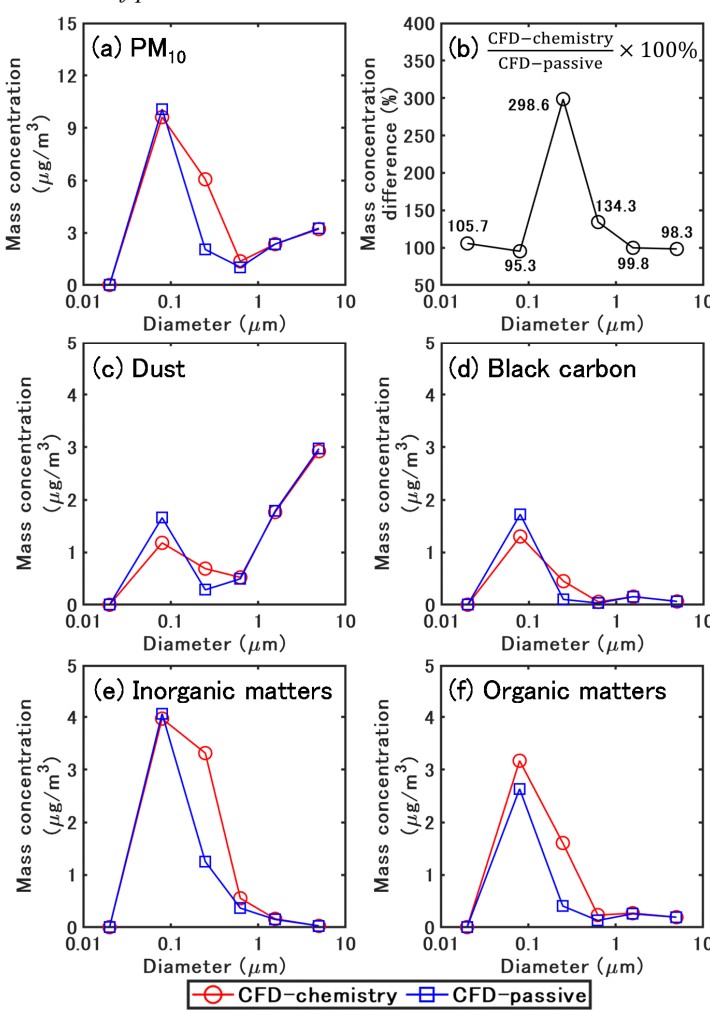


Fig. 17 Time-averaged size distribution of PM$_{10}$ for different chemical species from 5 a.m.
to 5 p.m.

Fig. 17 shows the time-averaged size distribution of PM$_{10}$ for the different chemical
compounds of particles from 5 a.m. to 5 p.m. The bound diameters are 0.01, 0.04, 0.16,
0.4, 1.0, 2.5 and 10 μm, and the mean diameters are 0.02, 0.08, 0.25, 0.63, 1.58 and 5.01
μm.
For the total concentration of PM$_{10}$ (Fig. 17(a)), the lowest and the largest concentrations
are in the first size section (0.01-0.04 μm) and the second size section (0.04-0.16 μm)





respectively, for both the CFD-passive and the CFD-chemistry simulations. Generally,
the loss and gain of mass concentration in each size section are related to emission, dry
deposition, coagulation (small particles coagulate into large particles), and
condensation/evaporation (phase exchange between gas and particles).
Fig. 17(b) shows the mass concentration ratio between CFD-passive and CFD-chemistry
for each size section. For particles in the size range of 0.04-0.16 μm, the concentrations
are smaller in CFD-chemistry than in CFD-passive, because dry deposition and
coagulation both decrease mass concentration for those particles. Furthermore, semi-
volatile gases may evaporate from small particles because of the Kelvin effect and
condense onto larger particles. For particles in the size range of 0.16-1.0 μm, the
concentrations are much larger in CFD-chemistry than CFD-passive, indicating that
coagulation and condensation on the mass-concentration increase are dominant to other
processes, such as deposition. For particles larger than 1 μm, the concentrations of CFD-
passive and CFD-chemistry were similar, because particle dynamics have a low influence
on large particles.
The size distribution of dust (Fig. 17(c)) shows that most dust mass concentrations were
in particles larger than 1 μm. Meanwhile, most of the mass concentration of BC, inorganic
and organic matters (Fig. 17(d-f)) is in particles smaller than 1 μm. Coagulation is the
main process influencing the size distribution for inert matters (dust and BC). Compared
to CFD-passive, the mass concentration of dust and BC in the second size section
decreased by 0.48 and 0.43 μg/m$^3$ in CFD-chemistry. Correspondingly, the mass
concentrations of dust and BC in the third size section increase by 0.41 and 0.35 μg/m$^3$.
For inorganic matters, in the second size section, the concentrations are similar in CFD-
passive and CFD-chemistry: particle dynamics decrease sulphate concentration by 0.32
μg/m$^3$ and increase nitrate concentration by 0.17μg/m$^3$. However, as the results of the
combination effect of coagulation and ammonium nitrate condensation, the
concentrations largely increased in the third size section in CFD-chemistry: sulphate,
ammonium and nitrate increased by 0.27, 0.6 and 1.24 μg/m$^3$, respectively.
For organic matters, because of condensation of hydrophilic compounds from background
biogenic gases and anthropogenic emissions, CFD-chemistry leads to a small increase in
concentrations (0.53 μg/m$^3$) in the second size section and a large increase in the third
section (1.21 μg/m$^3$) compared to CFD-passive. In detail, Bio concentrations increase by
0.89 μg/m$^3$, and Ant concentrations decrease by 0.36 μg/m$^3$ in the second size section. In
the third size section, Bio and Ant concentrations increase by 0.67, 0.54 μg/m$^3$.



*5.4. Influence of ammonia traffic emissions*

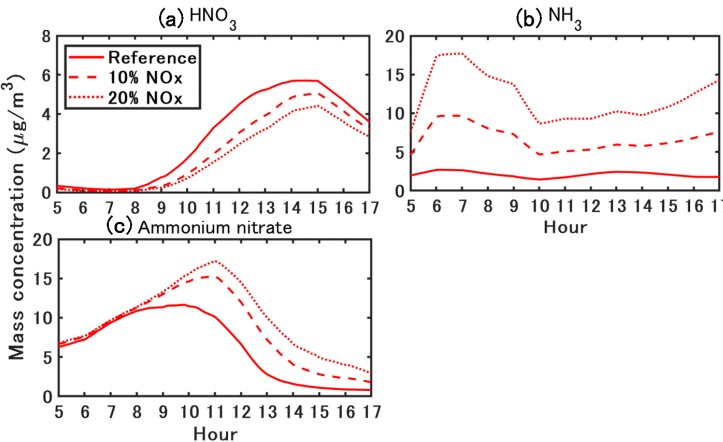


Fig. 18 Sensitivity of NH$_3$ emission on ammonium nitrate concentration.

Suarez-Bertoa et al., (2017) conducted on-road measurements of NH$_3$ emissions from two
Euro 6b compliant light duty cars (one gasoline and one diesel) under real-world driving
conditions, and they found that NH$_3$ emissions accounted for 11.9% and 0.92% of NOx
emissions for gasoline and diesel vehicles. As explained in Section 5.2, NH$_3$ emission
emissions were approximately 1-2% of NOx emissions in the reference case. Two cases
were considered to simulate the impact of an increase in the fraction of gasoline cars, and
sensitivity simulations were performed with NH$_3$ emission considered as 10% and 20%
of the NOx emissions.
Fig. 18 shows the sensitivity of ammonium nitrate concentration to NH$_3$ emissions. A
larger NH$_3$ emission delayed the peak of ammonium nitrate by approximately one hour.
For a 12-hour average, considering NH$_3$ emissions of 10% and 20% of NOx emissions
leads to a large increase in ammonium nitrate (35% and 55%) compared to the reference
case, because of the formation of ammonium nitrate by the condensation of HNO$_3$ and
NH$_3$.

**6. Conclusions**
Particles in urban environment impose adverse impacts on pedestrians' health.
Conventional CFD methods regarding particles as passive scalars cannot reproduce the
formation of secondary aerosols and may lead to uncertain simulations. Therefore, to
increase the simulation accuracy of particle dispersion, we coupled the CFD software
OpenFOAM (OF) and Code_Saturne (CS) with SSH-Aerosol, a modular box model to





simulate the evolution of primary and secondary aerosols. The main processes involved
in the aerosol dynamics (coagulation, condensation /evaporation and dry deposition) were
considered.
We simulated a 12-hour transient dispersion of pollutants from traffic emissions in a street
canyon using the unsteady RANS model. The simulation domain and background
concentration were based on field measurements. The flow field is based on the WRF
model. The particle diameter range (0.01 μm to 10 μm) was divided into six size sections.
The following conclusions were drawn from the results of this study.
1)  The simulated spatially-averaged values in the street canyon were validated from
field measurement using validation metrics. For both OF and CS, the simulated $NO_2$
and $PM_{10}$ concentrations based on the coupling model (CFD-chemistry) achieved
better agreement with the measurement data than the conventional CFD simulation
which considers pollutants as passive scalars (CFD-passive). The differences
between of the OF and CS results are not obvious and are mainly due to the
differences in the turbulence scheme. The following conclusions were drawn based
on the simulated OF concentrations.
2)  For the flow field, a large vortex was observed in the canyon with a small secondary
vortex at the corner of the leeward wall at the current aspect ratio ($H/W$=0.31). In
CFD-chemistry, because of the reverse flow, the 12-hour (from 5 a.m. to 5 p.m.) time-
averaged $NO_2$ mass concentration, $PM_{10}$ mass and number concentrations at
pedestrian height were much higher near the leeward wall (116 μg/m³, 28 μg/m³,
$3.2 \times 10^{10}$ m$^{-3}$) than the background (26 μg/m³, 15 μg/m³, $2.3 \times 10^{10}$ m$^{-3}$).
3)  Secondary aerosol formation largely affects the mass concentration and size
distribution of particle matters. For 12-hour time-averaged concentrations,
ammonium nitrate and organic matters increased by 46% and 53% in CFD-chemistry
compared to CFD-passive because of condensation of $HNO_3$ and $NH_3$, background
biogenic precursor-gases and anthropogenic precursor-gas emissions. Coagulation
largely influenced the size distribution of small particles by combining particles with
a diameter of 0.04-0.16 μm into 0.16-0.4 μm. At the same time, CFD-chemistry
showed a much larger concentration than CFD-passive for the particles in 0.16-1.0
μm, indicating that the effect of condensation on increasing mass concentration was
dominant compared to other chemical processes.
4)  Urban areas are $NH_3$-limited ($HNO_3$ sufficient) areas, therefore, increasing $NH_3$ leads
to a large increase in ammonium nitrate. Vehicles are considered to be the main
source of $NH_3$ in urban environments. Increasing the fleet's proportion of recent
gasoline vehicles may increase $NH_3$ emissions. For a 12-hour average, we considered


NH$_3$ emissions of 10% and 20% of NOx emissions led to a large increase in
ammonium nitrate (35% and 55%) compared to the reference case which considers
NH$_3$ emission as 1-2% of NOx emissions.

5)  A grid sensitivity analysis showed that the particles' concentrations of inorganic and
organic compounds were sensitive to grid resolution, whereas inert particle
concentrations were not sensitive to grid resolution. In addition, simulated values
based on a grid size of 0.5 m in the street canyon showed small differences with a
grid size of 0.25 m, indicating that a spatial resolution of 0.5 m can be enough for
reactive particle dispersion at the street level.

6)  Operator splitting is often employed to solve the transport term and chemical
reactions over a given time step in chemical transport simulations. Two integration
orders were considered: first order method (CFD($\Delta t$)-Chemistry($\Delta t$)) and Strang
method (CFD($\Delta t/2$)-Chemistry($\Delta t$)-CFD($\Delta t/2$)). The results showed that the Strang
method had almost the same concentrations as the first order method with half the
computational time. Further sensitivity analysis on the time step showed that a time
step of 0.5 s was enough when using the Strang method.

7)  Conducting a CFD simulation with constant boundary conditions and emission rates
at a specific time point is considered a practical method to achieve time-averaged
concentrations for evaluating street-level pollutant concentrations. The validation
was conducted using conditions on five time points (7 a.m., 10 a.m., 1 p.m., 3 p.m.
and 5 p.m.). The simulated concentration based on the above method exhibited
almost the same value as the simulation with transient conditions at the same time
points.

Future work will be conducted on the influence of environmental factors and emission
conditions, aiming to provide knowledge to devise suitable countermeasures to decrease
particle concentration in microscale urban environments.

Acknowledgments: This work benefited from discussions with Bertrand Carissimo. The
authors acknowledge funding from DIM QI[2] (Air Quality Research Network on air
quality in the Île-de-France region) and from Île-de-France region.

Code/Data availability
The codes used in this publication are available to the community, and they can be
accessed by request to the corresponding author.

Author contribution





KS and RO were responsible for conceptualization. CL, YW, CF, YK and ZW developed
the software. CL and YW conduced the visualization and validation; CL, YW and KS
performed the formal analysis. KS, YK and RO acquired resources. CL, YW, RO and KS
were responsible for writing and original draft preparation. CF, YK, HK reviewed and
edited the manuscript All co-authors contributed to the discussion of the paper.
Competing interests
The contact author has declared that neither they nor their co-authors have any
competing interests.

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

Appendix A
The schemes for particle deposition velocity $v_d$ were added to the transport equations
using volume sink terms based on Zhang et al. (2001) and can be represented as:

$$v_{d,p} = \begin{cases} v_g + \dfrac{1}{R_a + R_s}, & \text{Wall surfaces} \\ v_g, & \text{Entire field} \end{cases} \tag{A1}$$

$$v_g = \frac{\rho d_p^2 g C}{18\eta} \tag{A2}$$

$$R_a = \frac{\ln(z_R/z_0) - \psi_H}{\kappa u_*} \tag{A3}$$

$$R_s = \frac{1}{\varepsilon_0 u_* (E_B + E_{IM} + E_{IN}) R_1} \tag{A4}$$





The deposition velocity for the particles $v_{d,p}$ consists of both gravitational settling and
surface deposition near the wall surfaces. The gravitational settling velocity $v_g$ was
considered for the entire field, $\rho$ is the particle density; $d_p$ is the particle diameter; $g$
is the acceleration of gravity; $C$ is Cunningham correction factor for small particles; $\eta$
is the viscosity coefficient of air.
The aerodynamic resistance $R_a$ is calculated from the first-layer-height $z_R$, roughness
length $z_0$, Von Karman constant $\kappa$, friction velocity $u_*$ and stability function $\psi_H$. For the
k–ε model, $u_*$ is estimated by $\left(C_\mu^{0.5} k\right)^{0.5}$ and $C_\mu = 0.09$ is a constant of the model.
The surface resistance $R_s$ is calculated from $u_*$, the collection efficiency from Brownian
diffusion $E_B$, the impaction $E_{IM}$ and the interception $E_{IN}$. The correction factor
represents the fraction of particles that stick to the surface $R_1$ and an empirical
constant $\varepsilon_0 = 3$.
The dry deposition schemes for gas were added to the transport equations using volume
sink terms based on Wesely (1989) and Zhang et al. (2003),which can be represented as:

$$v_{d,g} = \frac{1}{R_a + R_b + R_c} \tag{A1}$$

$$R_b = \frac{2}{\kappa u_*}\left(\frac{Sc}{Pr}\right)^{2/3} \tag{A2}$$

The deposition velocity for gas $v_{d,g}$ is calculated from the aerodynamic resistance $R_a$,
the quasi-laminar layer resistance $R_b$ and the surface resistance for gas $R_c$. $Sc = v/D$
and $Pr = 0.72$ are the Schmidt and Prandtl number. $v$ is the kinematic viscosity of air
and $D$ is the molecular diffusivity of different gases. $R_c$ is calculated based on Zhang
et al. (2003).