# Peer review of "Modelling of street-scale pollutant dispersion by coupled simulation of"

_Atmospheric Chemistry and Physics, 2022_

## Referee Comment (RC3)

This ms examines coupled aerosol-chemistry-dynamics within a street canyon using a chemistry-aerosol box model, SSH-Aerosol, and two CFD models, OpenFOAM and Code_Saturne. The model configurations are intended to mimic conditions within a specific street canyon in Paris during a field campaign in 2014. It is shown that the coupled model leads to significant differences compared to a baseline simulation in which chemical and physical transformations are excluded.

Given that the vast majority of the urban pollutant dispersion literature assumes passive scalar dynamics, there is certainly a need for more studies of coupling, especially in the case of aerosols, which have received much less attention than gas-phase photochemistry. Nevertheless, several such studies have already appeared, e.g. Sanchez et al. (2016) [1], Han et al. 2018 [2], Gao et al. 2022 [3]. The authors should therefore highlight the novel aspects of their study. They should also discuss the physical or chemical mechanisms underlying their findings. While many sensitivity tests and models comparisons are presented, I did not get the impression that the ms attempts to address a basic physical question.

I also have several technical concerns/questions.

Major points

1. The CFD models are rather idealised.

    (a) I have doubts about the restriction to a 2-D street canyon. While it's well-established that flow within a street canyon is approximately two-dimensional, at least when the external flow is perpendicular to the canyon axis, this does not necessarily mean that a 2-D model works equally well for coupled simulations. Chemical and physical transformations depend on the residence time, which may differ for a fully three-dimensional flow. My hunch is that the residence time for a 3-D canyon would be shorter. Thus the effects of chemical reaction or aerosol processes could be weaker than what the authors report.

    (b) A RANS closure may not be sufficiently accurate. While RANS models agree well with large-eddy simulation when it comes to mean statistics, discrepancies are typically larger for second-order quantities. I suspect that there is a similar phenomenon for pollutants governed by nonlinear evolution equations.

    The authors should discuss the limitations of their models more carefully. How would their major findings be affected if a more realistic or accurate model (e.g. 3-D LES) were used?

2. The CFD models are not validated. This means that the results of, e.g., the model evaluation (Sec. 4), need to be interpreted carefully. On account of nonlinearity, the differences between CFD-chemistry and CFD-passive may not be due only to chemistry or aerosol processes. For example, the limitations discussed in Point 1 could also play an important role.

3. I don't understand how the time-dependent inflow is implemented.

    (a) It's mentioned that flow is driven by linearly interpolated time series of friction velocity and temperature (Figure 2), but how exactly is this done? Is there a time-dependent boundary condition? If so, how often is it updated? Are the authors sure that this procedure is robust?

    (b) Is the temperature a prognostic variable? It's unclear whether the simulations are neutral or not. An inlet velocity profile is specified (eq. 1) but the temperature profile is unspecified. If neutral stability is assumed, what's the justification for this?

(c) The specification of the velocity profiles ignores the dependence on wind direction. The authors assume that the external wind is always perpendicular to the canyon axis. For an application to a real street canyon, this could be a major omission.

(d) The description of the operator splitting seems incorrect. The authors claim that the symmetric Strang scheme [4], $CFD(\Delta t/2) - Chemistry(\Delta t) - CFD(\Delta t/2)$ is second order while the simple alternation of CFD and chemistry, $CFD(\Delta t) - Chemistry(\Delta t)$, is first order. There are a few problems with this.

    i. Strang's result actually requires that the numerical schemes for the decomposed sub-steps be second order (see p. 510 of his paper). The point isn't that the composition increases the order but rather maintains it. This is reasonable. Strang's method can be applied to general PDEs of the form $u_t = a + b$, where $a$ and $b$ are functions. If $a$ and $b$ are independent of each other but discretised using a first-order scheme, it seems rather unlikely that the composition should be of higher order.

    ii. Strang's result is for a PDE that's been decomposed into functions of independent variables, e.g. $u_t = a(Du, x, t) + b(Du, y, t)$, where $x$ and $y$ are the usual space variables and $Du$ represents spatial derivatives. The authors have equated the terms $a$ and $b$ with the chemistry and CFD substeps. I don't think this is the same thing. They're applying the method to a PDE of the form $u_t = a(Du_{CFD}, x, y, t) + b(u_{chem}, x, y, t)$ where the *dependent variables* $u_{CFD}$ and $u_{chem}$ denote subsets of the original solution space (i.e. $u = u_{CFD} + u_{chem}$). The operator splitting considered by the authors is conventional in the numerical solution of turbulent reacting flows, but it's different in kind from the splitting analysed by Strang (which generalises alternating direction methods).

    iii. I think it's misleading to imply that a higher-order scheme necessarily yields a more accurate solution of a turbulent reacting flow. The splitting should depend on the coupling between dynamics and chemistry, i.e. the Damkohler number. Thus a method, like the authors', which has a chemistry sub-timestep that's longer than the dynamical sub-timestep, will not be accurate in certain limits (e.g. slow chemistry) regardless of what the overall order might be.

    iv. If the Strang scheme is in fact higher-order or more accurate than the simple first-order scheme, then why should the simulated concentrations be largely insensitive to the timestep (Fig. 10)? This would imply that the actual errors are not described by classical error analysis, which is based on linearisation. If so, there's little point in referring to the order of the schemes.

4. The results are limited to a 12-hour simulation for a single day. Why? The authors should show that their results are robust by considering different days.

Minor points

1. The English should be checked carefully. There are many small errors. I suggest that the singular be used in place of 'inorganic and organic matters', etc.

2. The organisation of the ms could be improved. Here are several examples:

(a) The Abstract is too long and jumps between paragraphs.

(b) In the Introduction, the role of vehicles in emitting ammonia is mentioned after the OpenFOAM and Code_Saturne are introduced.

(c) Grid details aren't provided with the model description (Sec. 2) but deferred to Sec. 4.4

3. l.121 Why are two CFD models considered? One would expect well-tested models to give similar results. Why is the model comparison of general scientific interest?

4. l.133 Why is deposition implemented in the CFD models rather than SSH-aerosol? Since the chemistry and dynamics timesteps differ, this could introduce an inconsistency between deposition and other aerosol processes.

5. l.143 Why isn't nucleation considered?

6. l.144 Why were these bound diameters chosen?

7. l.173 Fig. 2 I presume that these figures correspond to a specific day. Why was this day chosen? Measurements were conducted from 6 April to 15 June.

8. l.191 ff. How is the emission spectrum of the aerosols defined?

9. l.204 What is the justification for setting the pressure to zero at the outlet? This choice could affect the flow within the canyon if the domain isn't sufficiently large. The authors should probably test the sensitivity to the domain size and/or perform a validation against wind-tunnel data.

10. l.263 Table 1 Some of the errors for CFD-chemistry are rather small. In light of the idealised numerical configuration, this seems fortuitous. It appears that the statistics correspond to a canyon average (i.e. Fig. 4). However, measurements are limited to a few points. Is this a fair comparison?

11. l.306 "This method is called the constant-condition method (CCM) in this study, in contrast to the transient-condition method (TCM)." As mentioned above, how is the transient-condition method implemented? Note that this information should be included in the model description.

12. l.318 "However, it should be noted that CCM cannot replace TCM when simulating long periods because the mass concentration may not change linearly between the selected time points." I don't understand this.

    (a) I think the issue isn't the length of the simulations but rather the representativeness of the inlet boundary conditions vis-a-vis the actual inflow.
    (b) Why is the mass concentration the only issue? In practice, variations in the inflow wind speed and direction play a crucial role.

References

1. Sanchez, B., J.-L. Santiago, A. Martilli, M. Palacios, and F. Kirchner, 2016: CFD modeling of reactive pollutant dispersion in simplified urban configurations with different chemical mechanisms. *Atmospheric Chemistry and Physics*, 16, 12143–12157.

2. Han, B.-S., J.-J. Baik, K.-H. Kwak, and S.-B. Park, 2018: Large-eddy simulation of reactive pollutant exchange at the top of a street canyon. *Atmospheric Environment*, 187, 381–389.

3. Gao, S., M. Kurppa, C. K. Chan, and K. Ngan, 2022: Technical note: Dispersion of cooking-generated aerosols from an urban street canyon. *Atmospheric Chemistry and Physics*, 22, 2703–2726.

4. Strang, G., 1968: On the Construction and Comparison of Difference Schemes, *SIAM J. Numer. Anal.*, 5(3), 506–517.

---

## Author Comment (AC1)

**Response to Comments**

Dear Reviewers:

We truly appreciate your comments on our manuscript entitled "Modelling of street-scale pollutant dispersion by coupled simulation of chemical reaction, aerosol dynamics, and CFD" (Manuscript Number: acp-2022-365). Your comments are valuable and have been very helpful in improving our paper. We have carefully studied the comments and made the appropriate corrections, which we hope will meet your approval. In this document, the reviewers' comments have been written in blue, while our answers have been typed in black. All changes have been highlighted in red in the revised manuscript. The responses to the reviewers' comments are presented below.

**Response to the reviewers' comments**

Reviewer #1: This paper used a coupled simulation system to study the $NO_2$ and aerosol distributions in a street canyon at Paris, which is of some interest and fits into the scope of ACP. However, there are still some deficiencies in the article, and a minor revision is needed before publication at ACPD:

First of all, it is found that the simulation domain is rather small, i.e. only a street canyon. The chemical concentrations measured or simulated are significantly influenced by the outside conditions, either meteorological or chemical boundary conditions. Therefore, it is necessary to verify the conditions of simulation results outside the domain to make sure that they are realistic and provide the real boundary conditions for the street simulations. In this study, the focus was put on the pollutants' physical phenomenon inside the street canyon, therefore the simulation domain was relatively small. For the inflow, the meteorological boundary conditions including the time variations of the friction velocity and temperature were obtained from the WRF model. And the time variations of the background concentrations were obtained from a regional chemistry-transport model as described in Sartelet et al. (2018). Gas chemistry and aerosol dynamics were considered in the regional model, with the same chemical representation as in this study. As detailed in Sartelet et al. (2018), the simulated regional concentrations compare well to measurements of $O_3$, $NO_2$, $PM_{10}$, $PM_{2.5}$, black carbon and organic aerosols. Therefore, the authors considered the boundary conditions were suitable and real for the street simulations. The related statement was added at Line 221.

The comparisons between model simulation and field measurements were largely done for $NO_2$ and aerosol species. It is suggested that the comparisons be done for wind velocities and directions between model simulation and observations. As many conclusions drawn from the paper were based on the flow field simulated, such as shown

in Fig. 12-13, it must make sure that the flows simulated were realistic.

Thank you for your advice. The authors agree that reproducing the flow field is important in this study. Unfortunately, we do not have the observation data on wind velocity. Therefore, we conducted a velocity validation for OpenFOAM v2012 using data from a wind tunnel experiment (Blackman et al., 2015).

[Figure]

Fig. S1 Simulation domain for velocity validation.

The 2-D simulation domain is shown in Fig. S1. The aspect ratio in the experiment (H/W=0.33) is close to this study (H/W=0.31). The building height $H$ is 0.06 m. The grid size is 1/20 H in $x$- and $z$- directions in the simulation domain under $3H$. The free-stream velocity $U_{ref}$ is 5.9 m/s. The steady-state flow field is simulated with the same turbulence model (RNG k–ε model) as in the paper, and cyclic boundary conditions are used for the inlet and outlet. The slip boundary is considered for the top, and non-slip boundary conditions with the same wall functions as in the paper are considered for other walls.

[Figure]

Fig. S2 Streamwise and vertical direction of mean wind velocities at $z/H = 0.83$.

Fig. S2 compares the simulated streamwise and vertical direction of mean wind velocities with the experimental values at $z/H = 0.83$. The RNG k–ε model reproduce well the velocities, although the velocities very close to the windward wall show differences with the experimental values. The above validation shows that if suitable inlet conditions are given, the flow field is well reproduced with the turbulence model adopted in this study. As mentioned above, measurement velocity data are not available, therefore the authors

could only deduce that the flows simulated were realistic based on the validation using wind tunnel experiment data. The velocity validation was added in Appendix B and the related statement was added at the beginning of Section 4.1.

Since chemical reactions are the major concerns of this paper, the street level variations of temperature, flow patterns et al, should be discussed with respect to the formation of $O_3$ and aerosol dynamics.

The authors agree that the radiation-leaded street-level-variations of temperature could affect the flow field and chemical reaction rates. However, this is not considered in this study and the radiation effect on the local temperature was simplified as being the same as in the inflow condition. The inflow temperature was obtained from WRF model where the radiation was considered, and the time variation of temperature was considered to be the same as the background. Further work will consider the implementation of the radiation effect. The above statements were added in the Conclusions as the limitation of this study.

The authors agree that the chemical formation and aerosol dynamics should be discussed with respect to the flow patterns and added the following sentence at Line 460. "In addition, the lowest concentration of $O_3$ and $HNO_3$ can be found at the leeward corner, which corresponds to the secondary vortex in Fig. 11, indicating that the pollutant residence time is the highest at that corner leading to enhance ozone titration."

Reference for Reviewer #1

Sartelet, K., Zhu, S., Moukhtar, S., André, M., André, J. M., Gros, V., Favez, O., Brasseur, A. and Redaelli, M.: Emission of intermediate, semi and low volatile organic compounds from traffic and their impact on secondary organic aerosol concentrations over Greater Paris, Atmos. Environ., 180, 126–137, doi:10.1016/j.atmosenv.2018.02.031, 2018.
Blackman, K., Perret, L. and Savory, E.: Effect of upstream flow regime on street canyon flow mean turbulence statistics, Environ. Fluid Mech., 15(4), 823–849, doi:10.1007/s10652-014-9386-8, 2015.

Reviewer #2: The authors coupled the SSH-Aerosol modular box model into Code_Saturne and OpenFOAM CFD models. This coupling allows the representation of primary and secondary pollutants such as NO2 and secondary aerosols (including size distribution) inside an urban canyon in Paris. Results of twelve-hour simulations of $NO_2$ and $PM_{10}$ between OpenFOAM and Code_Saturne with SSH-Aerosol were similar and closer to measurements. Configuration experiments and case studies were focused on OpenFOAM coupling (CFD_Chemistry). CFD-Chemistry was evaluated in grid size, time step, and coupling method. Averaged concentration fields of gasses and particles and the size distribution of PM10 were evaluated. Finally, a case increasing $NH_3$ traffic emission was presented as an example of using the model for regulatory purposes.
 The manuscript is well written and organized, the tables and figures support the results and the results and conclusions represent an improvement in the field.
Minor comments and required clarifications are detailed below:

Specific comments

Line 143. Please clarify the reason for not including nucleation in simulations.
Nucleation is not considered in this study because only the mass and not the number of particles is available for evaluation, and large uncertainties remain on the nucleation parameterizations (Sartelet et al. 2022), mostly affecting the number of particles. As nucleation is not considered, the minimum diameter does not need to be as low as 0.001 μm, and it is fixed to 0.01 μm, as in the regional-scale simulations of Sartelet et al., (2018), which provide the background concentrations. The above explanation was added at Line 154.

Line 146 - 153. Recommend adding a schematic diagram to clarify the coupling detailed in this paragraph, like Fig 3 in Kim et al (2018).
The authors considered that the coupling method between CFD and chemistry modules is similar to the literature (Kurppa et al., 2019; Gao et al., 2022), and therefore a schematic diagram is not necessary. A related statement was added at the beginning of Section 2.

Fig 3. could also include NO2 and O3 background concentrations that would complement the discussion of Fig. 12. It could also help to have the gasses emission rates like NOX as it was also used to estimate NH3 emissions.
The time variations of $NO_2$ and $O_3$ background concentrations were added in Fig 3(a). The time variations of $NO_2$ and NO emission rates were added in Fig 3(b).

Line 177. Please specify the spatial resolution of WRF simulations.
The grid resolution was 1 km ×1 km in Paris. The statement was added at Line 202.

Line 193 Please specify the model name to produce the background concentrations.

The simulations are carried out with the Eulerian model Polair3D of the Polyphemus air quality modelling platform (Mallet et al., 2007). The statement was added at Line 224.

Line 241-243 Fig. 4.b showed that the PM10 CFD-chemistry did improve PM10 simulation but only during the morning hours, CFD-chemistry is closed to CFD-passive during the afternoon. When compared with Fig 3 a. It seems that it follows the behavior of background concentration, even when there is an increase of PM10 emitted compounds during the late hour of the afternoon (Fig 3 b). Did the authors perform sensitive tests of background concentration?

The reason why CFD-chemistry was closed to CFD-passive during the afternoon could be that the temperature increased, and relative humidity decreased during the day, therefore the low humidity in the afternoon decreased condensation rate. The authors considered that the reason for underestimating $PM_{10}$ in CFD-chemistry could be underestimation of background concentrations and underestimation of emission for black carbon as shown in Fig. 5.

Line 293: Please elaborate on the reasons for choosing OpenFOAM instead of Code_Saturne. Maybe it is easier and faster to run or it performs better than Code_Saturne (Table 1 only showed performance statistics for OpenFOAM). In that sense, it is also important to show the difference in computational time of running CFD-Chemistry and CFD-Passive.

Because the simulation results based on OpenFOAM and Code_Saturne were close, the authors considered that it is arbitrary to select one or the other CFD software in this study. OpenFOAM was selected without special reason, and the authors considered that the results based on OpenFOAM are representative. In addition, the simulation time ratio of CFD-chemistry and CFD-passive is about three times in both OpenFOAM and Code_Saturne. The statement was added at Line 281.

Line 591 - 592. The authors said that background concentration came from measurements, nevertheless in line 193 they said that they were obtained from the regional-scale simulations, please clarify.

"The simulation domain and background concentration were based on field measurements" was revised into "The simulation domain was generated to model a street-canyon where field measurements are available. The background concentrations of gas and particles are obtained from regional-scale simulations." at Line 629.

Technical corrections

Line 124 and Line 127. Please include the definition of RNG (Re-Normalisation Group), PISO (Pressure Implicit with Splitting of Operator), and SIMPLE (Semi-Implicit Method for Pressure-Linked Equations) acronyms.

The definitions were added at Lines 133 and 136.

Recommend to add units in colorbars in Fig. 12 and 13, like in Fig. 11.
The units were added in Fig. 12 and 13.

Line 466, It should be: the HNO3 concentration peak in CFD-chemistry was later than that in CFD-PASSIVE.
Revised at Line 504.

Line 458 Said NH3 concentration peaked around 6 am, later in line 466 is said to be at 7 am.
Revised 6 a.m. to 7 a.m. at Line 497.

Line 505, and line 508 PM10 are repeated, maybe it is OM.
In the caption of Fig.16, two $PM_{10}$ stand for the total concentration of $PM_{10}$ and the concentration for individual chemical compound of $PM_{10}$. To avoid ambiguity, we changed the order into $PM_{10}$, $PM_1$ and the chemical compounds of $PM_{10}$ in the caption of Fig.16 and Line 546.

Line 834, von Karman constant should be von Kármán constant
Revised at Line 909.

Equations for dry deposition schemes for gasses in Appendix A should be labeled as A5 and A6.
Revised.

Reference for Reviewer #2
Sartelet, K., Kim, Y., Couvidat, F., Merkel, M., Petäjä T., Sciare J. and Wiedensohler, A. (2022), Influence of emission size distribution and nucleation on number concentrations over Greater Paris. Atmos. Chem. Phys., 22, 8579-8596, doi:10.5194/acp-22-8579-2022
Sartelet, K., Zhu, S., Moukhtar, S., André, M., André, J. M., Gros, V., Favez, O., Brasseur, A. and Redaelli, M.: Emission of intermediate, semi and low volatile organic compounds from traffic and their impact on secondary organic aerosol concentrations over Greater Paris, Atmos. Environ., 180, 126–137, doi:10.1016/j.atmosenv.2018.02.031, 2018.
Kurppa, M., Hellsten, A., Roldin, P., Kokkola, H., Tonttila, J., Auvinen, M., Kent, C., Gao, S., Kurppa, M., Chan, C. K. and Ngan, K.: Technical note: Dispersion of cooking-generated aerosols from an urban street canyon, Atmos. Chem. Phys., 22(4), 2703–2726, doi:10.5194/acp-22-2703-2022, 2022.
Mallet, V., Quélo, D., Sportisse, B., Ahmed De Biasi, M., Debry, E., Korsakissok, I., Wu, L., Roustan, Y., Sartelet, K., Tombette, M., and Foudhil, H.: Atmospheric Chemistry and Physics Technical Note: The air quality modeling system Polyphemus, Atmos. Chem. Phys, 7, 5479–5487, 2007.

Reviewer #3: This ms examines coupled aerosol-chemistry-dynamics within a street canyon using a chemistry-aerosol box model, SSH-Aerosol, and two CFD models, OpenFOAM and Code_Saturne. The model configurations are intended to mimic conditions within a specific street canyon in Paris during a field campaign in 2014. It is shown that the coupled model leads to significant differences compared to a baseline simulation in which chemical and physical transformations are excluded.

Given that the vast majority of the urban pollutant dispersion literature assumes passive scalar dynamics, there is certainly a need for more studies of coupling, especially in the case of aerosols, which have received much less attention than gas-phase photochemistry. Nevertheless, several such studies have already appeared, e.g. Sanchez et al. (2016) [1], Han et al. 2018 [2], Gao et al. 2022 [3]. The authors should therefore highlight the novel aspects of their study. They should also discuss the physical or chemical mechanisms underlying their findings. While many sensitivity tests and models comparisons are presented, I did not get the impression that the ms attempts to address a basic physical question.

I also have several technical concerns/questions.

The authors agree that this study shared similar motivation with previous studies on developing a coupling method between CFD and chemistry modules. Meanwhile, the authors reviewed the previous studies on developing a coupling method between CFD and aerosol modules in the introduction and considered that the gas-phase chemistry and secondary (organic) aerosol formation were not fully considered. Therefore, this study aims to develop a more comprehensive coupled model which can simulate the evolution of gas concentrations, mass and number concentrations of primary and secondary particles at the same time. The above explanation was added at Line 80.

The authors agree that discussing the physical or chemical mechanisms based on the simulation results is important. Meanwhile, the focus of this study is to develop and evaluate the coupled model of CFD and chemistry modules. Following studies will focus on addressing basic physical questions based on the developed model. However, a first evaluation of the physical and chemical mechanisms is presented here to evaluate the role of ammonia emissions in the formation of inorganic and organic aerosols. As the hydrophilicity of organic aerosols is considered in the model here, it is shown that ammonia does not only impact inorganic concentrations, but also organic concentrations.

Major points
1. The CFD models are rather idealised.
(a) I have doubts about the restriction to a 2-D street canyon. While it's well-established that flow within a street canyon is approximately two-dimensional, at least when the external flow is perpendicular to the canyon axis, this does not necessarily mean that a 2-D model works equally well for coupled simulations. Chemical and physical transformations depend on the residence time, which may differ for a fully threedimensional flow. My hunch is that the residence time for a 3-D canyon would be shorter. Thus the effects of chemical reaction or aerosol processes could be weaker than what the authors report.

The authors agree that the pollutant residence time for a 3-D canyon would be shorter compared to the 2-D canyon adopted in this study. Meanwhile, the selected street canyon in this study is simple in geometry and a 2-D simplification is reasonable in terms of residence time for a perpendicular wind, as shown by the 2D-3D comparisons performed by Maison et al. (2022). Nevertheless, the authors agree that the 3-D characteristics of the flow field are important in simulating pollutant dispersion with chemical reactions when the wind direction is time dependent. The above statement was added in the Conclusions as a limitation of this study.

(b) A RANS closure may not be sufficiently accurate. While RANS models agree well with large-eddy simulation when it comes to mean statistics, discrepancies are typically larger for second-order quantities. I suspect that there is a similar phenomenon for pollutants governed by nonlinear evolution equations.
The authors should discuss the limitations of their models more carefully. How would their major findings be affected if a more realistic or accurate model (e.g. 3-D LES) were used?

The authors agree that RANS models may show discrepancies with LES for second-order quantities. As the SSH-aerosol processed the ensemble-averaged concentration, the covariance of turbulent diffusion and chemical reaction may not be fully reproduced. The above statement was added in the Model description (Line 168) and was added in the Conclusions as a limitation of this study.

2. The CFD models are not validated. This means that the results of, e.g., the model evaluation (Sec. 4), need to be interpreted carefully. On account of nonlinearity, the differences between CFD-chemistry and CFD-passive may not be due only to chemistry or aerosol processes. For example, the limitations discussed in Point 1 could also play an important role.

The authors agree that model evaluation for CFD is important in this study. Meanwhile, we do not have the observation data on wind velocity. Therefore, we conducted a velocity validation for OpenFOAM using data from a wind tunnel experiment (Blackman et al., 2015). The velocity validation was added as Appendix B and a related statement was added at the beginning of Section 4.1.

3. I don't understand how the time-dependent inflow is implemented.
(a) It's mentioned that flow is driven by linearly interpolated time series of friction velocity and temperature (Figure 2), but how exactly is this done? Is there a time-dependent boundary condition? If so, how often is it updated? Are the authors sure that this procedure is robust?

The boundary conditions for inflow and background concentrations are time-dependent.

The time series from the WRF model and the regional model for each hour is interpolated into values for each time step in the CFD. The authors consider this procedure is reliable. The following statement was added at Line 214. "The hourly friction velocities and temperatures are linearly interpolated into seconds and prescribed at the inflow."

(b) Is the temperature a prognostic variable? It's unclear whether the simulations are neutral or not. An inlet velocity profile is specified (eq. 1) but the temperature profile is unspecified. If neutral stability is assumed, what's the justification for this?
Since the domain height is low (51 m) in this study and we focused on the pollutant dispersion behaviors in the street canyon, it is reasonable to consider the atmospheric stability as neutral, therefore the temperature was spatially uniform at the inflow. The above statement was added at Line 211.

(c) The specification of the velocity profiles ignores the dependence on wind direction. The authors assume that the external wind is always perpendicular to the canyon axis. For an application to a real street canyon, this could be a major omission.
The authors agree that various wind directions should be considered to better evaluate the performance of the coupled model in 3D cases. Meanwhile, since we focused on the coupling of gas chemical reactions and particle dynamics to the CFD codes, we selected a period when the wind direction was perpendicular to the street canyon. Further work will focus on the application of the coupled model to a complex urban environment with changing wind directions. The above explanations were added in the conclusions as a limitation of this study.

(d) The description of the operator splitting seems incorrect. The authors claim that the symmetric Strang scheme [4], $CFD(\Delta t/2) - Chemistry(\Delta t) - CFD(\Delta t/2)$ is second order while the simple alternation of CFD and chemistry, $CFD(\Delta t) - Chemistry(\Delta t)$, is first order. There are a few problems with this.
i. Strang's result actually requires that the numerical schemes for the decomposed substeps be second order (see p. 510 of his paper). The point isn't that the composition increases the order but rather maintains it. This is reasonable. Strang's method can be applied to general PDEs of the form $u_t = a + b$, where a and b are functions. If a and b are independent of each other but discretised using a first-order scheme, it seems rather unlikely that the composition should be of higher order.
ii. Strang's result is for a PDE that's been decomposed into functions of independent variables, e.g. $u_t = a(Du, x, t) + b(Du, y, t)$, where x and y are the usual space variables and Du represents spatial derivatives. The authors have equated the terms a and b with the chemistry and CFD substeps. I don't think this is the same thing. They're applying the method to a PDE of the form $u_t = a(Du_{CFD}, x, t) + b(u_{chem}, y, t)$ where the dependent variables $u_{CFD}$ and $u_{chem}$ denote subsets of the original solution space (i.e. $u = u_{CFD} + u_{chem}$). The operator splitting considered by the authors is conventional in the numerical solution of turbulent reacting flows, but it's different in kind from the

splitting analysed by Strang (which generalizes alternating direction methods).
iii. I think it's misleading to imply that a higher-order scheme necessarily yields a more accurate solution of a turbulent reacting flow. The splitting should depend on the coupling between dynamics and chemistry, i.e. the Damkohler number. Thus a method, like the authors', which has a chemistry sub-timestep that's longer than the dynamical sub-timestep, will not be accurate in certain limits (e.g. slow chemistry) regardless of what the overall order might be.
iv. If the Strang scheme is in fact higher-order or more accurate than the simple first-order scheme, then why should the simulated concentrations be largely insensitive to the timestep (Fig. 10)? This would imply that the actual errors are not described by classical error analysis, which is based on linearisation. If so, there's little point in referring to the order of the schemes.

The authors appreciate the detailed explanation on the Strang method and agree that the operator splitting method of $CFD(\Delta t/2) - Chemistry(\Delta t) - CFD(\Delta t/2)$ is conventional in the numerical solution of turbulent reacting flows and is different to the Strang method. In addition, the authors agree that the above method is first-order and even a higher-order scheme would not be accurate in certain limits (e.g. slow chemistry). Therefore, the statements on the Strang method were revised. The name of first-order method and the Strang method were revised into A-B splitting method and A-B-A splitting method. The Section 4.4 was revised correspondingly.

4. The results are limited to a 12-hour simulation for a single day. Why? The authors should show that their results are robust by considering different days.

The authors agree that conducting simulation for different days may bring a more comprehensive evaluation of the coupled model. Meanwhile, as answered above, as a first step in the evaluation of the coupled model, we focused on the reproduction of gas chemical reactions and particle dynamics in a street canyon. We selected the time period when wind direction was perpendicular to the street, so that a 2-D simplification of the simulation domain is reasonable. During the field measurements, there exists several time periods when the wind direction was perpendicular to the street canyon, but the duration of these time periods were most of the time very too short. In addition, the authors consider that it is critical to have a simulation time long enough to cover both day-time chemistry and night-time chemistry. Therefore, we selected the 12-hour period on April 30, 2014. Further work will focus on the application of the coupled model to different days. The reason for selecting the simulation period was added at Line 187. And the corresponding statement were added in the conclusions as a limitation of this study.

Minor points
1. The English should be checked carefully. There are many small errors. I suggest that the singular be used in place of 'inorganic and organic matters', etc.

Thank you for the suggestion. The authors revised 'matters' into 'matter'. In addition, the tense in the manuscript was revised.

2. The organisation of the ms could be improved. Here are several examples:
(a) The Abstract is too long and jumps between paragraphs.
The Abstract was shorten into single paragraphs.

(b) In the Introduction, the role of vehicles in emitting ammonia is mentioned after the OpenFOAM and Code_Saturne are introduced.
The role of vehicles in emitting ammonia was put before the introduction of the CFD softwares at Line 83.

(c) Grid details aren't provided with the model description (Sec. 2) but deferred to Sec. 4.4
The following statements were added in Section 3 (Line 183). "The grid resolutions in the street canyon are 0.5 m in both x-and z- directions, respectively. The largest grid sizes are 4 m (x) × 2m (z)."

3. l.121 Why are two CFD models considered? One would expect well-tested models to give similar results. Why is the model comparison of general scientific interest?
The authors considered that both OpenFOAM and Code_Saturne own wide users. Therefore, coupling SSH-aerosol with both CFD softwares may satisfy more needs. The above explanation was added at Line 95. Furthermore, there may be uncertainties linked to the numerical resolutions used in the CFD codes. This study shows that these uncertainties do not affect the simulated concentrations when the CFD model is coupled to an aerosol module.

4. l.133 Why is deposition implemented in the CFD models rather than SSH-aerosol? Since the chemistry and dynamics timesteps differ, this could introduce an inconsistency between deposition and other aerosol processes.
The authors consider that the deposition is related with walls in CFD and is similar with diffusion. In addition, the chemistry and dynamics timesteps were the same in A-B splitting. And the simulation results based on A-B-A splitting were close with A-B splitting. Therefore, the authors considered that deposition implemented in the CFD models is more reasonable rather than SSH-aerosol.

5. l.143 Why isn't nucleation considered?
Nucleation is not considered in this study because only the mass and not the number of particles is available for evaluation, and large uncertainties remain on the nucleation parameterizations (Sartelet et al. 2022), mostly affecting the number of particles. As nucleation is not considered, the minimum diameter does not need to be as low as 0.001 μm, and it is fixed to 0.01 μm, as in the regional-scale simulations of Sartelet et al., (2018), which provide the background concentrations. The above explanation was added at Line 154.

6. l.144 Why were these bound diameters chosen?

6. l.144 Why were these bound diameters chosen?

The bound diameters were chosen to be the same as those of the background concentrations from the regional-scale simulations of Sartelet et al., (2018). The statement was added at Line 156.

7. l.173 Fig. 2 I presume that these figures correspond to a specific day. Why was this day chosen? Measurements were conducted from 6 April to 15 June.

As answered above, the authors selected a time period when the wind direction was perpendicular to the street for a relatively long period (12-hour time period).

8. l.191 How is the emission spectrum of the aerosols defined?

The PM size distribution at emission is assumed to be the same as in authors' previous study (Lugon et al., 2021 a, b), i.e. exhaust primary PM is assumed to be in the size bin [0.04 – 0.16 μm] while non-exhaust primary PM is coarser in the size bin [0.4 – 10 μm]. The statement was added at Line 237.

9. l.204 What is the justification for setting the pressure to zero at the outlet? This choice could affect the flow within the canyon if the domain isn't sufficiently large. The authors should probably test the sensitivity to the domain size and/or perform a validation against wind-tunnel data.

The authors agree that the zero-pressure setting may affect the downstream flow development if the domain is not sufficiently large. Meanwhile, the authors considered that the distance from the street canyon to the outlet, which was about 3 times of building height, was long enough for flow development at the downstream roof. In addition, although the outlet setting in velocity validation was different with this study, the prediction accuracy was confirmed.

10. l.263 Table 1 Some of the errors for CFD-chemistry are rather small. In light of the idealized numerical configuration, this seems fortuitous. It appears that the statistics correspond to a canyon average (i.e. Fig. 4). However, measurements are limited to a few points. Is this a fair comparison?

In the field measurement, the measured concentration was obtained from averaging over two measurement points at the leeward and windward walls at different heights. The measured concentrations shown in this study are these averaged values Therefore, the authors considered that these measured concentrations could be represented by the canyon average value. A related statement was added at Line 274.

11. l.306 "This method is called the constant-condition method (CCM) in this study, in contrast to the transient-condition method (TCM)." As mentioned above, how is the transient-condition method implemented? Note that this information should be included in the model description.

For TCM, the inflow conditions, pollutants' background concentrations and emission rates from regional models were linearly interpolated into each time step. The authors moved the introduction of CCM and TCM from Section 4.2 to the beginning of the model description.

12. l.318 "However, it should be noted that CCM cannot replace TCM when simulating long periods because the mass concentration may not change linearly between the selected time points." I don't understand this.
(a) I think the issue isn't the length of the simulations but rather the representativeness of the inlet boundary conditions vis-a-vis the actual inflow.
(b) Why is the mass concentration the only issue? In practice, variations in the inflow wind speed and direction play a crucial role.
The authors agree with the reviewer and revised the short comings of CCM as the followings at Line 356." However, CCM should be use with caution when the inflow wind speed and direction vary rapidly."

References from Reviewer #3
1. Sanchez, B., J.-L. Santiago, A. Martilli, M. Palacios, and F. Kirchner, 2016: CFD modeling of reactive pollutant dispersion in simplified urban configurations with different chemical mechanisms. Atmospheric Chemistry and Physics, 16, 12143–12157.
2. Han, B.-S., J.-J. Baik, K.-H. Kwak, and S.-B. Park, 2018: Large-eddy simulation of reactive pollutant exchange at the top of a street canyon. Atmospheric Environment, 187, 381–389.
3. Gao, S., M. Kurppa, C. K. Chan, and K. Ngan, 2022: Technical note: Dispersion of cooking-generated aerosols from an urban street canyon. Atmospheric Chemistry and Physics, 22, 2703–2726.
4. Strang, G., 1968: On the Construction and Comparison of Difference Schemes, SIAM J. Numer. Anal., 5(3), 506–517.

References for Reviewer #3
Blackman, K., Perret, L. and Savory, E.: Effect of upstream flow regime on street canyon flow mean turbulence statistics, Environ. Fluid Mech., 15(4), 823–849, doi:10.1007/s10652-014-9386-8, 2015.
Uhrner, U., von Löwis, S., Vehkamäki, H., Wehner, B., Bräsel, S., Hermann, M., Stratmann, F., Kulmala, M. and Wiedensohler, A.: Dilution and aerosol dynamics within a diesel car exhaust plume-CFD simulations of on-road measurement conditions, Atmos. Environ., 41(35), 7440–7461, doi:10.1016/j.atmosenv.2007.05.057, 2007.
Sartelet, K., Zhu, S., Moukhtar, S., André, M., André, J. M., Gros, V., Favez, O., Brasseur, A. and Redaelli, M.: Emission of intermediate, semi and low volatile organic compounds from traffic and their impact on secondary organic aerosol concentrations over Greater Paris, Atmos. Environ., 180, 126–137, doi:10.1016/j.atmosenv.2018.02.031, 2018.
Lugon, L., Vigneron, J., Debert, C., Chrétien, O., and Sartelet, K.: Black carbon

modeling in urban areas: investigating the influence of resuspension and non-exhaust emissions in streets using the Street-in-Grid model for inert particles (SinG-inert), Geoscientific Model Development, 14, 7001–7019, https://doi.org/10.5194/gmd-14-7001-2021, 2021a.

Lugon, L., Sartelet, K., Kim, Y., Vigneron, J., and Chretien, O.: Simulation of primary and secondary particles in the streets of Paris using MUNICH, Faraday Discussions, 226, 432–456, https://doi.org/10.1039/d0fd00092b, 2021b.

Maison, A., Flageul, C., Carissimo, B., Wang, Y., Tuzet, A., and Sartelet, K. (2022), Parameterizing the aerodynamic effect of trees in street canyons for the street network model MUNICH using the CFD model Code_Saturne. Atmos. Chem. Phys., 22, 9369-9388, doi:10.5194/acp-22-9369-2022

Sartelet, K., Kim, Y., Couvidat, F., Merkel, M., Petäjä T., Sciare J. and Wiedensohler, A. (2022), Influence of emission size distribution and nucleation on number concentrations over Greater Paris. Atmos. Chem. Phys., 22, 8579-8596, doi:10.5194/acp-22-8579-2022

---

## Author Response (AR2)

**Response to Comments**

Dear Reviewer:

We truly appreciate your comments on our manuscript entitled "Modelling of street-scale pollutant dispersion by coupled simulation of chemical reaction, aerosol dynamics, and CFD" (Manuscript Number: acp-2022-365). Your comments are valuable and have been very helpful in improving our paper. We have carefully studied the comments and made the appropriate corrections, which we hope will meet your approval. In this document, the reviewers' comments have been written in blue, while our answers have been typed in black. All changes have been highlighted in red in the revised manuscript. The responses to the reviewers' comments are presented below.

**Response to the reviewers' comments**

Reviewer #1:
The main concern for this paper is the methodology used in simulating the street level pollutants.
(1) The street domain is too small and ideal, not representing the real street configuration. Therefore, a detailed evaluation of the modeling results cannot be carried out.
In this study, the focus was put on the pollutants' physical phenomenon inside the street canyon, therefore the simulation domain was relatively small. The authors consider that the 2-D simplification of the street canyon is reasonable for a perpendicular wind, as shown by the small concentration difference in the 2D-3D comparisons performed by Maison et al. (2022). In addition, the 2-D simplification is frequently adopted for studying dispersion of reactive pollutants in a street canyon (Garmory et al., 2009; Wu et al., 2021). The related statements were added in Line 695.

(2) For such a small scale CFD simulations, the dynamical (3-D wind speed), thermal (Temperature) and mass (PM, BC, NO2,..) boundary conditions are the critical time-varying inputs to the CFD model, but the study failed to specify clearly.
The authors agree that the boundary conditions are the critical time-varying inputs to the CFD model. In this study, the hourly friction velocities, temperatures, and background concentrations at the inlet were from the regional model and were linearly interpolated into each timestep. The background concentrations were spatial-uniformly prescribed at the inflow and top boundaries. Therefore, the general trends were simulated but the fast fluctuations at the inlet were not reproduced. Nevertheless, this method is frequently adopted in the RANS-based simulations of street-level pollutant dispersion (Kim et al., 2019, Kumar et al., 2009). The related statements were added in Line 215 and 231.

(3) For the pollutants, the street level concentrations are composed by two contributions: local streel emissions and transports from outside the domain. If the outside

contributions were not specified and validated correctively, the modeling results from this study cannot be sufficiently evaluated.

As detailed in Sartelet et al. (2018), the simulated regional concentrations compare well to measurements of $O_3$, $NO_2$, $PM_{10}$, $PM_{2.5}$, black carbon and organic aerosols. Therefore, the authors considered the boundary conditions were suitable and real for the street simulations. The related statements can be found in Line 229.

Reference

Maison, A., Flageul, C., Carissimo, B., Wang, Y., Tuzet, A., and Sartelet, K.: Parameterizing the aerodynamic effect of trees in street canyons for the street network model MUNICH using the CFD model Code_Saturne, Atmos Chem Phys, 22, 9369–9388, https://doi.org/10.5194/acp-22-9369-2022, 2022.

Garmory, A., Kim, I. S., Britter, R. E. and Mastorakos, E.: Simulations of the dispersion of reactive pollutants in a street canyon, considering different chemical mechanisms and micromixing, Atmos. Environ., 43(31), 4670–4680, doi:10.1016/j.atmosenv.2008.07.033, 2009.

Wu, L., Hang, J., Wang, X., Shao, M. and Gong, C.: APFoam 1.0: Integrated computational fluid dynamics simulation of O3-NOx-volatile organic compound chemistry and pollutant dispersion in a typical street canyon, Geosci. Model Dev., 14(7), 4655–4681, doi:10.5194/gmd-14-4655-2021, 2021.

Kim, M. J., Park, R. J., Kim, J. J., Park, S. H., Chang, L. S., Lee, D. G., and Choi, J. Y.: Computational fluid dynamics simulation of reactive fine particulate matter in a street canyon, Atmos Environ, 209, 54–66, https://doi.org/10.1016/j.atmosenv.2019.04.013, 2019.

Kumar, P., Garmory, A., Ketzel, M., Berkowicz, R., Britter, R.: Comparative study of measured and modelled number concentrations of nanoparticles in an urban street canyon, Atmos Environ, 43, 949-958, https://doi.org/10.1016/j.atmosenv.2008.10.025, 2009.

Sartelet, K., Zhu, S., Moukhtar, S., André, M., André, J. M., Gros, V., Favez, O., Brasseur, A. and Redaelli, M.: Emission of intermediate, semi and low volatile organic compounds from traffic and their impact on secondary organic aerosol concentrations over Greater Paris, Atmos. Environ., 180, 126–137, doi:10.1016/j.atmosenv.2018.02.031, 2018.